# Beyond Ensembles: Simulating All-Atom Protein Dynamics in a Learned Latent Space

Aditya Sengar[*][‡]
aditya.sengar@epfl.ch

Jiying Zhang[*][†]
jiying.zhang@epfl.ch

Pierre Vandergheynst[*]
pierre.vandergheynst@epfl.ch

Patrick Barth[†][§][‡]
patrick.barth@epfl.ch

## Abstract

Simulating the long-timescale dynamics of biomolecules is a central challenge in computational science. While enhanced sampling methods can accelerate these simulations, they rely on pre-defined collective variables that are often difficult to identify, restricting their ability to model complex switching mechanisms between metastable states. A recent generative model, LD-FPG, demonstrated that this problem could be bypassed by learning to sample the static equilibrium ensemble as all-atom deformations from a reference structure, establishing a powerful method for all-atom ensemble generation. However, while this approach successfully captures a system's probable conformations, it does not model the temporal evolution between them. We introduce the Graph Latent Dynamics Propagator (GLDP), a modular component for simulating dynamics within the learned latent space of LD-FPG. We then compare three classes of propagators: (i) score-guided Langevin dynamics, (ii) Koopman-based linear operators, and (iii) autoregressive neural networks. Within a unified encoder–propagator–decoder framework, we evaluate long-horizon stability, backbone and side-chain ensemble fidelity, and temporal kinetics via TICA. Benchmarks on systems ranging from small peptides to mixed-topology proteins and large GPCRs reveal that autoregressive neural networks deliver the most robust long rollouts and coherent physical timescales; score-guided Langevin best recovers side-chain thermodynamics when the score is well learned; and Koopman provides an interpretable, lightweight baseline that tends to damp fluctuations. These results clarify the trade-offs among propagators and offer practical guidance for latent-space simulators of all-atom protein dynamics.

## 1 Introduction

Molecular simulations are indispensable for studying the complex dynamics that govern biological function, yet brute-force approaches struggle to access the slow, functionally relevant motions—such as protein folding, ligand binding, or allosteric switching—due to rugged energy landscapes and the dominance of rare events (Dror et al., 2011; Latorraca et al., 2017). To mitigate this gap, beyond what enhanced sampling can offer when suitable collective variables are hard to specify (Chen, 2021), a complementary strategy has gained traction: recasting the simulation challenge from one of brute-force integration to one of representation learning. In this *representation-first* view, the simulation problem is recast as a modular encoder–propagator–decoder pipeline: an encoder maps high-dimensional atomic configurations into a continuous, low-dimensional latent space; a propagator evolves the system's state within this simplified space; and a decoder maps the resulting latent trajectory back to all-atom coordinates (Sidky et al., 2020; Wu et al., 2018).

Progress in this paradigm follows two complementary directions: The first focuses on learning the underlying physics, employing score-based diffusion (Song et al., 2020; Ho et al., 2020), flow

[*]Signal Processing Laboratory (LTS2), EPFL, Lausanne, Switzerland
[†]Institute of Bioengineering, EPFL, Lausanne, Switzerland
[‡]Corresponding author
[§]Ludwig Institute for Cancer Research, Lausanne, Switzerland

matching (Lipman et al., 2022), and energy-based models (LeCun et al., 2006) to learn generative surrogates (sometimes yielding differentiable force fields) that implicitly define the system's potential of mean force (Arts et al., 2023; Plainer et al., 2025; Hsu et al., 2024; Li et al., 2024). The second centers on learning simplified dynamical coordinates, using time-aware autoencoders or Koopman/DMD analysis (Lusch et al., 2018; Tu, 2013) to discover an intrinsic manifold where the long-term dynamics become stable, predictable, or even approximately linear (Hernández et al., 2018; Mardt et al., 2018; Brunton et al., 2021; Azencot et al., 2020; Ghorbani et al., 2022).

Despite the promise of latent-space simulation, the choice of propagator (the engine that advances the state) entails a trade-off between physical faithfulness, long-horizon stability, and expressive capacity. We adopt LD-FPG (Sengar et al., 2025) as a fixed encoder–decoder and compare, in the same latent, three propagators: (i) score-guided Langevin dynamics, (ii) a Koopman linear operator, and (iii) an autoregressive neural network (NN). Our analysis reveals a clear trade-off: Autoregressive NN is most stable, the score-guided model is most physically detailed for local motions, and the Koopman operator is a simple but overly rigid baseline.

GLDP is designed as a system-specific surrogate for MD, akin to LSS, GeoTDM, ITO (Sidky et al., 2020; Han et al., 2024; Schreiner et al., 2023). The objective is to learn the specific transition operator of a target system to enable accelerated sampling of its equilibrium and kinetics. While cross-protein generalization (as pursued by foundation models) is a valuable complementary goal, this work focuses on maximizing thermodynamic fidelity and long-horizon stability for specific, complex targets.

**Contributions:** (1) A controlled study that holds the LD-FPG encoder–decoder fixed while swapping the latent propagator, isolating where gains arise. (2) Three propagators within one latent: score-guided Langevin, a Koopman operator, and an autoregressive NN. (3) Benchmarks on alanine dipeptide, 7JFL, HIV-1 Protease (1R6W), and $A_1AR$. (4) First learned, all-atom rollout (to our knowledge) that recovers a GPCR activation surface ($A_2AR$).

## 2 RELATED WORK

**Latent simulators and encoder–propagator–decoder pipelines.** Simulating dynamics in a low-dimensional latent space has been explored in several molecular simulators. Molecular Latent Space Simulators (LSS) (Sidky et al., 2020) and Deep Generative MSMs (DeepGenMSM) (Wu et al., 2018) already factorize the task into an encoder that maps coordinates to latent variables, a propagator that advances them, and a decoder that returns all-atom structures. Follow-up work uses recurrent propagators (Vlachas et al., 2021), trajectory-level generators that synthesize MD clips in coordinate space (Jing et al., 2024), and LSS variants that target free-energy surfaces and transition kinetics (Dobers et al., 2023). Joint conformation–dynamics models trained via autoregression have also been proposed (Shen et al., 2025). Our framework follows this modular design but differs in two respects: (i) we freeze the LD-FPG encoder–decoder (Sengar et al., 2025) to control reconstruction quality and (ii) we perform a focused comparison of different propagators within the *same* latent, isolating how the propagation rule affects stability, ensemble fidelity, and functional observables.

**Learning dynamically aware latent spaces.** The quality of a latent simulator is largely determined by its encoder. Classical TICA is a standard tool for discovering slow coordinates (Pérez-Hernández et al., 2013), while deep time-lagged autoencoders such as VDE learn nonlinear, delay-predictive representations (Hernández et al., 2018; Wehmeyer & Noé, 2018). VAMPnets approximate leading Koopman eigenfunctions to capture the slowest processes (Mardt et al., 2018), and more recent work uses GNNs to learn symmetry-aware collective variables directly from coordinates, sometimes with information-bottleneck objectives to optimize predictiveness of future states (Zou et al., 2025; Ghorbani et al., 2022; Zhang et al., 2024; Wang & Tiwary, 2021); see Chen & Chipot (2023) for a survey. In this work we instead keep the LD-FPG ChebNet encoder fixed (Sengar et al., 2025) and treat its latent as given, focusing on how different propagation mechanisms behave once a reconstruction-ready latent space has been trained.

**Propagators: linear Koopman, neural sequence models, and stochastic dynamics.** The Koopman-operator view approximates nonlinear dynamics with a linear map in a learned observable space (Brunton et al., 2021), with EDMD/DMD estimating this map directly (Williams et al., 2015; Tu, 2013) and deep variants learning the observables so that latent evolution is linear (Lusch et al.,

2018; Azencot et al., 2020; Tayal et al., 2023; Nayak et al., 2025); implicit transfer-operator learning further parameterizes Markov kernels with diffusion models under SE(3) symmetry (Schreiner et al., 2023). A second line directly fits nonlinear transition maps using neural sequence models, including LSTMs/RNNs for effective dynamics (Vlachas et al., 2021), RNNs with Maximum Caliber constraints (Tsai et al., 2022), and continuous-time Neural ODE/SDE formulations (Chen et al., 2018; Kidger et al., 2021), with ConfRover combining an autoregressive temporal module and an SE(3)-equivariant diffusion decoder (Shen et al., 2025). A third class uses stochastic dynamics guided by generative models: score-based diffusion linked to Langevin-type simulators and effective force fields (Song et al., 2020; Arts et al., 2023; Hsu et al., 2024), and flow matching for coarse-grained rollouts and free-energy modeling (Li et al., 2024; Kohler et al., 2023), often with Fokker–Planck-inspired objectives to align equilibrium and stationary laws (Plainer et al., 2025). Our three propagators Koopman, autoregressive NN, and score-guided Langevin, are representative of these families but are all evaluated inside a single, fixed latent space.

**Coordinate-space generative simulators and MD surrogates.** Many recent models operate directly in Cartesian or SE(3)-equivariant coordinate space: GeoTDM uses diffusion for trajectory synthesis (Han et al., 2024), $F^3$low applies flow matching to coarse-grained rollouts (Li et al., 2024), EGNO targets multi-step 3D dynamics with equivariant neural operators (Xu et al., 2024), and NeuralMD couples group-symmetric differential equations with multi-resolution protein–ligand dynamics (Liu et al., 2024). Geometric diffusion models such as GeoDiff and DiffDock focus on static structures and poses (Xu et al., 2022; Corso et al., 2022), while methods like LAST, ConfRover, and NeuralMD use learned representations to steer or unify MD-like dynamics (Tian et al., 2022; Shen et al., 2025; Liu et al., 2024). Our setting is closer to system-specific surrogates such as LSS, GeoTDM, and ITO (Sidky et al., 2020; Han et al., 2024; Schreiner et al., 2023): GLDP is trained per system to accelerate sampling of its equilibrium and kinetics, but keeps the all-atom decoder frozen and swaps only the propagator, enabling a controlled comparison of dynamical assumptions within a fixed latent representation.

**Benchmark systems for protein dynamics.** We probe kinetic fidelity across a hierarchy of systems, from small peptides to complex membrane receptors. Alanine dipeptide (ADP) is a canonical baseline with a well-characterized dihedral free-energy surface (Kalko et al., 1999; Vymetal & Vondrasek, 2010; Klein & Noé, 2024). At intermediate complexity, we use globular proteins 7JFL and HIV-1 protease (1R6W) from the ATLAS database, which provides standardized all-atom trajectories under common protocols (Vander Meersche et al., 2024). The most demanding tests are G protein-coupled receptors (GPCRs), specifically the adenosine $A_1$ and $A_{2A}$ receptors ($A_1AR$, $A_2AR$), whose function involves rare, slow transitions between metastable states (Dror et al., 2011; Latorraca et al., 2017) mediated by well-mapped microswitches and allosteric pathways (Rasmussen et al., 2011; Weis & Kobilka, 2018). Using recent high-quality MD datasets for these GPCRs (D'Amore et al., 2024), we test whether latent propagators can match both equilibrium observables and activation free-energy surfaces when the encoder and decoder are held fixed.

## 3 METHODS

GLDP extends the LD-FPG encoder–decoder backbone (Sengar et al., 2025) (see App. A.2) by adding a temporal propagator that operates entirely within the LD-FPG latent. The encoder–decoder is kept fixed (frozen) in all experiments; only the propagator is swapped.

### 3.1 GLDP: PROPAGATION IN THE LD-FPG LATENT SPACE

We adopt the LD-FPG encoder–decoder backbone (Sengar et al., 2025). The workflow follows a strict four-step sequence:

1. **Encoding:** A Chebyshev Graph Neural Network (ChebNet) maps all-atom coordinates $X(t) \in \mathbb{R}^{N \times 3}$ to per-atom embeddings.
2. **Pooling:** A deterministic pooling layer aggregates these embeddings into a single compact latent vector $z(t) \in \mathbb{R}^d$.
3. **Propagation:** The GLDP advances the latent state $z_t \to z_{t+1}$ using the learned transition function.

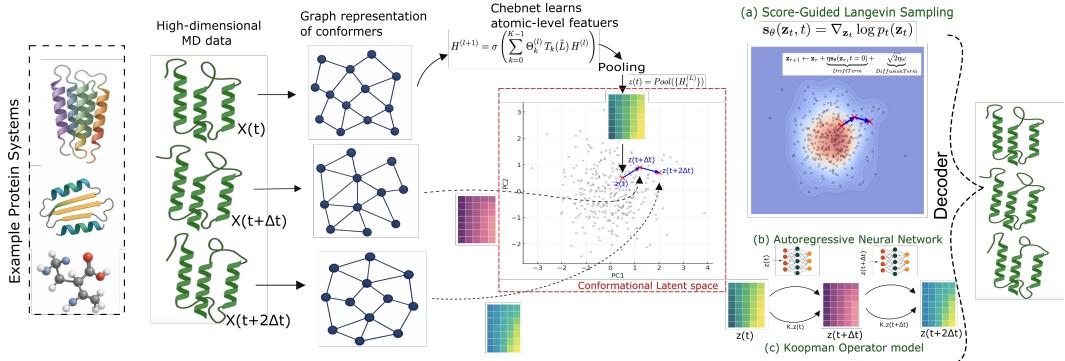

Figure 1: **Framework overview.** A pre-trained LD-FPG encoder (ChebNet; left) maps all-atom coordinates $X(t)$ to a pooled latent $z(t)$. Within the fixed LD-FPG latent, GLDP advances the state via one of three propagators (red box): **(a)** score-guided Langevin using the LD-FPG denoiser to estimate $s_\theta(z, \tau) = \nabla_z \log p_\tau(z)$ at a fixed low-noise level; **(b)** an autoregressive NN $z_{t+1} = f_\theta(z_t)$; and **(c)** a Koopman linear operator $z_{t+1} = A z_t$. The frozen LD-FPG decoder (right) maps the latent trajectory back to all-atom structures $\hat{X}(t + \Delta t)$.

4. **Decoding:** The fixed decoder maps the evolved latent $z_{t+1}$ back to all-atom coordinates $\hat{X}(t+1)$.

The encoder produces a time series $\{z_0, z_1, \dots, z_M\}$ that captures conformational variation. We then learn a latent-time update of the form

$$z_{t+1} = f(z_t) + \eta_t, \qquad \eta_t \sim \mathcal{N}(0, \sigma_\eta^2 I), \tag{1}$$

where $f$ is the propagator and $\eta_t$ is optional rollout noise. Unless noted, training pairs use a one-frame stride $(z_t, z_{t+1})$, latents are standardized per dimension using training-set statistics before fitting the propagator (and unstandardized before decoding), and the initial state $\hat{z}_0$ is the encoding of a held-out MD frame. Key notation appears in Table 14; a glossary is in Table 15.

## 3.2 KOOPMAN PROPAGATOR VIA DYNAMIC MODE DECOMPOSITION

The Koopman variant approximates latent evolution with a single linear operator:

$$z_{t+1} \approx A\, z_t, \qquad A \in \mathbb{R}^{d \times d}.$$

where $A$ is the time-independent Koopman operator that we aim to learn from data. We estimate $A$ with Dynamic Mode Decomposition (DMD) (Tu, 2013). First, we arrange our latent trajectory into two snapshot matrices, where each column is a state of the system at a point in time,

$$\mathbf{X} = [z_0, \dots, z_{M-2}], \qquad \mathbf{Y} = [z_1, \dots, z_{M-1}],$$

and solve the least-squares map $\min_A \|\mathbf{Y} - A\mathbf{X}\|_F^2$, yielding

$$A = \mathbf{Y}\,\mathbf{X}^+,$$

where $\mathbf{X}^+$ is the Moore–Penrose pseudoinverse. For numerical stability and to regularize the model, the pseudoinverse is computed using Singular Value Decomposition (SVD). By truncating the SVD to a lower rank $r$, we can filter out noise and focus the model on the most energetically significant dynamic modes. Once learned, the operator $A$ is used to generate new trajectories autoregressively from an initial state $\hat{z}_0$; a small Gaussian rollout noise $\eta_t \sim \mathcal{N}(0, \sigma_{\text{koop}}^2 I)$ is added to match the one-step residual variance. Implementation details are provided in App. A.3.

## 3.3 AUTOREGRESSIVE NEURAL NETWORK PROPAGATOR

To learn the latent transition directly, we use an autoregressive NN. The model parameterizes a deterministic nonlinear map

$$z_{t+1} = f_\theta(z_t), \tag{2}$$

trained with a one–step objective over pairs $(z_t, z_{t+1})$:

$$\mathcal{L}(\theta) \;=\; \frac{1}{M-1} \sum_{t=0}^{M-2} \left\| f_\theta(z_t) - z_{t+1} \right\|^2. \tag{3}$$

To reduce error accumulation in long rollouts, we use dropout and weight decay (see App. A.4 for architecture and training settings). After training, trajectories are generated by iterative application from an initial state $\hat{z}_0$:

$$\hat{z}_{i+1} \;=\; f_\theta(\hat{z}_i) \;+\; \eta_i, \qquad \eta_i \sim \mathcal{N}(0, \sigma_{\text{roll}}^2 I), \tag{4}$$

where $\sigma_{\text{roll}}$ is chosen on a validation split to match the one–step latent residual variance of $z_{t+1} - z_t$. (Unless noted, $\hat{z}_0$ is the encoding of a held-out MD frame and latents are standardized per dimension before fitting and unstandardized before decoding.) The network architecture, training procedure, and rollout settings are detailed in Appendix A.4.

### 3.4 Score-Guided Langevin Propagator

This propagator uses a learned score of the latent equilibrium to define a stochastic simulation. The procedure has two stages.

**Stage 1: score model.** We train a time-conditional denoiser $\epsilon_\theta(z_t, t)$ on latent embeddings with a variance-preserving (VP) schedule as in DDPMs. The forward corruption is

$$z_t \;=\; \sqrt{\bar{\alpha}_t}\, z_0 \;+\; \sqrt{1 - \bar{\alpha}_t}\, \epsilon, \quad \epsilon \sim \mathcal{N}(0, I), \quad \bar{\alpha}_t = \prod_{s=1}^{t}(1 - \beta_s).$$

When the network is trained to predict noise, the perturbed-data score is

$$s(z_t) \;=\; \nabla_z \log p_t(z_t) \;\approx\; -\frac{\epsilon_\theta(z_t, t)}{\sqrt{1 - \bar{\alpha}_t}}.$$

To approximate the *unperturbed* score $s(z)$ at test time, we query a low noise level $t_{\text{noise}} \approx 0$ (we use $t_{\text{noise}} = 0$ unless stated).

**Stage 2: Langevin simulation in latent space.** We integrate an overdamped Langevin SDE with Euler–Maruyama, using the score as the drift:

$$z_{t+1} \;=\; z_t \;+\; T\,\Delta t\, s(z_t) \;+\; \sqrt{2\,T\,\Delta t}\, \eta_t, \qquad \eta_t \sim \mathcal{N}(0, \sigma_\eta^2 I). \tag{5}$$

The effective diffusion variance per step is $2\,T\,\Delta t\,\sigma_\eta^2$; we set $\sigma_\eta = 1$ unless otherwise noted. Equivalently, if $p(z) \propto \exp(-U(z)/T)$, then $T\,s(z) = -\nabla U(z)$, so the drift corresponds to the gradient of an effective energy surface in latent space.

The temperature $T$, step size $\Delta t$, and other implementation details, such as score clipping and sampling stride are further described in Appendix A.5.

**Target ensemble.** Our propagators evolve only the latent coordinate $z$ (no momenta, no barostat/box dynamics). The objective is to reproduce the *configuration-space* equilibrium of the reference MD at its thermodynamic state. Let $p_X(x)$ denote the equilibrium distribution of all-atom coordinates $x$ from the source MD, and let $E[\cdot]$ and $D[\cdot]$ be the fixed LD-FPG encoder and decoder. The target latent law is the pushforward $p_Z = E_\# p_X$ with $z = E(x)$ and decoded samples are $\hat{x} = D(z)$. We fit propagators so that the stationary decoded trajectory $\{\hat{x}(t)\}$ matches reference statistics in $X$-space (e.g., dihedral histograms, contacts, and TM-distance free-energy surfaces). Because no Hamiltonian with momenta is integrated in latent space, we do not claim NVE energy conservation or exact physical kinetics. The score-guided Langevin propagator targets a canonical law in latent space whose invariant density is $p_Z$ when the score is exact; Koopman and the autoregressive NN model approximate the one-step transition kernel $K(z_{t+1} \mid z_t)$ observed from $(z_t, z_{t+1})$ under the fixed encoder. Unless stated otherwise, comparisons of timescales are diagnostic rather than calibrated to wall-clock time.

**Rationale for Latent Dynamics.** Simulating directly in high-dimensional Cartesian space ($3N$) is computationally demanding and highly sensitive to error accumulation, which often violates steric constraints. GLDP circumvents this by propagating dynamics on a compressed manifold ($d \ll 3N$). Because the LD-FPG latent space encodes deformations relative to a physical reference structure, the dynamics are inherently centered within a valid conformational basin. This effectively decouples the simulation challenge: the fixed *Decoder* enforces local geometric validity (bond lengths and angles), allowing the *Propagator* to focus exclusively on learning slow, collective motions on a smoothed energy landscape free of high-frequency noise.

## 4 RESULTS AND DISCUSSION

**Key result.** GLDP operates reliably across scales (ADP $\rightarrow$ 7JFL $\rightarrow$ 1R6W $\rightarrow$ $A_1AR$) and reproduces the activation surface of $A_2AR$; this cross scale pattern is the main message of the paper. See Appendix A.8 for $A_1AR$ ablations (encoder width, pooling size, and Langevin/denoiser settings).

We first position GLDP against recent generative simulators using GLDP-NN (GLDP with the autoregressive NN propagator)(Sec. 4.1). We then compare Koopman, NN, and Langevin within the same latent space on ADP and $A_1AR$ for stability and ensemble fidelity (Sec. 4.2, 4.3), and finally assess all three on $A_2AR$ functional surfaces (Sec. 4.4).

### 4.1 COMPARATIVE BENCHMARKS ON TRAJECTORY GENERATION

To benchmark GLDP, we compared it against a diverse set of state-of-the-art models: the conceptually similar **Latent Space Simulator (LSS)** (Sidky et al., 2020), the video-synthesis-based **MD-Gen** (Jing et al., 2024), and **GeoTDM** (Han et al., 2024), a leading coordinate-based diffusion model. All methods were evaluated on equilibrium fidelity, flexibility, and—crucially—temporal kinetics across systems of increasing complexity: the canonical **alanine dipeptide (ADP)**, the $\alpha$-helical **7JFL**, the mixed $\alpha/\beta$ **HIV-1 Protease (1R6W)**, and the functionally complex **$A_1AR$ GPCR**. The quantitative results are summarized in Table 1.[1]

Table 1: **Benchmark of dynamics models.** GLDP-NN is compared against LSS, MD-Gen, and GeoTDM. We include kinetic metrics (TICA timescales and decorrelation times) to assess temporal fidelity alongside structural ensemble metrics. Lower JSD is better. Higher Contact Correlation is better. For Kinetics, values closer to the Ground Truth are better. **Bold** indicates the best result among models. N/A: not applicable.

| System | Model | JSD | | Contact Corr. | Kinetics (steps) | |
| | | Dihedral (BB/SC) | Coord. (BB/SC) | | TICA ($t_1$) | Decorr. ($t_{decorr}$) |
|---|---|---|---|---|---|---|
| **ADP** | Ground Truth | N/A | N/A | N/A | 16.41 | 49 |
| | LSS | **0.016** / N/A | 0.621 / 0.633 | N/A | **15.80** | **46** |
| | GeoTDM | 0.123 / N/A | 0.444 / 0.395 | N/A | 12.20 | 35 |
| | GLDP | 0.061 / N/A | **0.366 / 0.386** | N/A | 10.14 | 28 |
| **7JFL** | Ground Truth | N/A | N/A | N/A | 660.34 | 2224 |
| | LSS | **0.027** / 0.097 | 0.691 / 0.682 | 0.935 | **610.50** | **2050** |
| | MD-Gen | 0.035 / 0.090 | **0.377 / 0.361** | **0.960** | 480.20 | 1600 |
| | GLDP | 0.037 / **0.081** | 0.426 / 0.459 | 0.949 | 226.40 | 850 |
| **1R6W** | Ground Truth | N/A | N/A | N/A | 1670.11 | 2207 |
| | LSS | **0.035** / 0.135 | 0.620 / 0.610 | 0.885 | 102.50 | 210 |
| | MD-Gen | 0.050 / 0.105 | **0.190** / 0.210 | **0.980** | 450.30 | 980 |
| | GLDP | 0.045 / **0.092** | 0.220 / **0.185** | 0.972 | **980.10** | **1650** |
| **$A_1AR$** | Ground Truth | N/A | N/A | N/A | 1058.85 | 1281 |
| | LSS | 0.134 / 0.146 | 0.657 / 0.602 | 0.971 | Unstable | Unstable |
| | MD-Gen | 0.033 / 0.088 | 0.106 / 0.117 | 0.941 | 85.40 | 150 |
| | GLDP | **0.019 / 0.067** | **0.063 / 0.087** | **0.985** | **650.20** | **520** |

---

[1]GeoTDM on 7JFL, 1R6W and $A_1AR$ exceeded 80 GB GPU memory of H100. MD-Gen does not support capped termini in our ADP setup; ADP results omitted.

**Ensemble Fidelity:** We assess equilibrium fidelity via Jensen–Shannon divergence (JSD) over global coordinates and local dihedrals. On the small ADP system, LSS leads on backbone dihedrals, while GLDP attains the best coordinate JSD. On 7JFL, results are split, with LSS and MD-Gen performing well on backbone metrics, while GLDP best captures side-chain packing. The introduction of **1R6W** challenges the models with a mixed topology; here, LSS captures local backbone dihedrals best and MD-Gen excels in backbone coordinate placement, yet GLDP achieves the lowest side-chain divergence (JSD 0.092 vs 0.135 for LSS) and superior side-chain coordinate fidelity (0.185). On the large $A_1AR$, GLDP dominates, achieving the lowest JSDs for both coordinates and dihedrals, indicating the most faithful recovery of tertiary structure in large, membrane-bound systems. While Table 1 quantifies distributional error via JSD, we provide a visual comparison of the backbone and side-chain free-energy landscapes for ADP and $A_1AR$ against baselines (LSS, MD-Gen, GeoTDM) in App. Figure 6– 8.

**Global contacts / long-horizon stability:** As a coarse proxy for structural stability, we compare the Pearson correlation between model and reference time-averaged contact maps. On 7JFL, MD-Gen best matches the contacts ($r = 0.960$), followed by GLDP ($r = 0.949$). This trend continues for the mixed-topology **1R6W**, where MD-Gen ($r = 0.980$) and GLDP ($r = 0.972$) both maintain excellent tertiary definition, whereas LSS degrades ($r = 0.885$), failing to preserve the $\beta$-sheet interface. However, on the most complex system, $A_1AR$, GLDP attains the highest correlation ($r = 0.985$), suggesting that while MD-Gen is strong on globular proteins, GLDP offers superior stability for complex transmembrane bundles.

**Temporal Kinetics (TICA):** To verify that models capture *dynamics* rather than just independent samples, we quantify the slowest TICA timescale ($t_1$) and decorrelation time ($t_{\mathrm{decorr}}$) (for more details check Appendix A.7). On simpler systems (ADP, 7JFL), baselines like LSS effectively recover slow modes. However, complexity reveals a sharp divergence. On **1R6W**, GLDP recovers a timescale of 980 steps and deep memory ($t_{\mathrm{decorr}} = 1650$), significantly outperforming the faster decorrelation of MD-Gen ($t_{\mathrm{decorr}} = 980$). Most notably on **$A_1AR$**, LSS exhibits numerical instability (negative eigenvalues), whereas GLDP remains physically robust ($t_1 \approx 650$ steps), proving it models coherent kinetics where baselines diverge.

**Summary.** GLDP offers the strongest balance of structural realism and dynamic fidelity. While baselines like MD-Gen or LSS can excel on specific static metrics for simpler topologies, GLDP consistently outperforms on side-chain packing and flexibility for complex systems like 1R6W and $A_1AR$. Crucially, the kinetic analysis confirms that GLDP is the only method stable enough to recover physically interpretable timescales on large GPCRs, positioning it as a highly dependable latent-space simulator.

## 4.2 LONG-HORIZON STABILITY: AUTOREGRESSIVE NN IS MOST ROBUST

Figure 2 tracks RMSD/lDDT[2] over rollout length and reports the *failure time* (first frame with lDDT $< 0.65$ relative to the initial frame). This threshold was empirically calibrated: in our analysis, trajectories dropping below 0.65 consistently exhibited unphysical degradation (e.g., secondary structure melting or steric clashes), distinguishing true model failure from valid conformational drift (where lDDT typically remains $> 0.70$). On alanine dipeptide, the autoregressive NN remains stable for the full 10,000-frame horizon without failure. The Koopman operator persists for thousands of frames (failure at 4,443), whereas Langevin terminates early (206) due to the high sensitivity of the score estimate on this small system. On $A_1AR$, the NN again completes the full 10,000-frame horizon without failure; Langevin reaches 7,476 and Koopman 5,740. Taken together, the NN is the most robust propagator for long-horizon rollouts on both a small peptide and a large GPCR, with Koopman providing a competitive linear baseline and Langevin exhibiting sensitivity to score/integration settings.

**What a "frame" means for Langevin (and how we calibrated it).** For Koopman and NN, one frame equals one dataset stride ($z_t \rightarrow z_{t+1}$), so the horizontal axis matches the MD sampling stride. The Langevin propagator integrates a latent SDE with internal step size $\Delta t$ and a separate sampling

---

[2]The local Distance Difference Test (lDDT) evaluates local structural quality by measuring the preservation of inter-atomic distances.

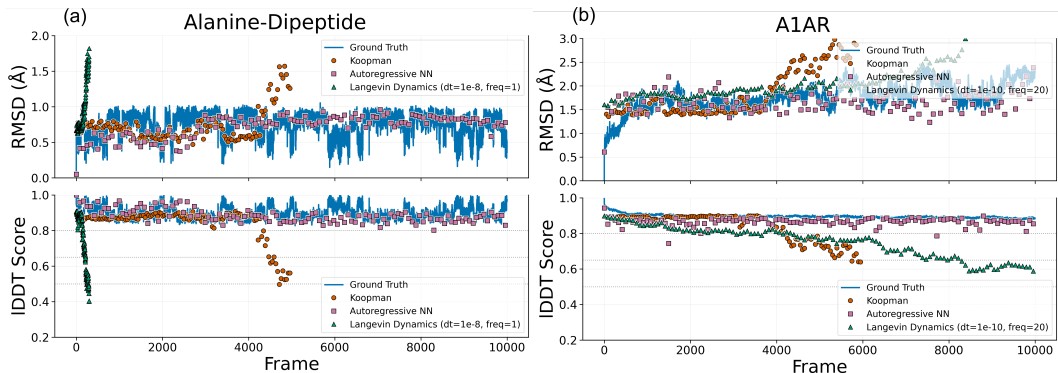

Figure 2: **Stability over long rollouts.** RMSD and lDDT versus frame index for (a) alanine dipeptide and (b) $A_1AR$. We define *failure time* as the first frame whose lDDT (relative to the initial frame) drops below 0.65. On $A_1AR$, the autoregressive NN remains stable for the entire 10,000-frame horizon (no failure), while Langevin and Koopman fail earlier; on alanine dipeptide, Koopman and NN persist for thousands of frames whereas Langevin fails early.

stride $s$; a plotted "frame" is a *sampled* SDE state, not a single integrator step. To make curves comparable, we calibrated $(\Delta t, s)$ to match the *short-horizon RMSD-per-frame* of the reference MD (Sec. A.5): for alanine, $\Delta t = 10^{-8}$ and $s{=}1$; for $A_1AR$, $\Delta t = 10^{-10}$ and $s{=}20$ (effective time per plotted frame $= s\,\Delta t$). Reported Langevin failure indices therefore count sampled outputs.

**Why Langevin fails earlier on alanine.** We attribute the early failure on alanine to three factors: (i) a sparsely sampled latent manifold from a large MD frame stride, yielding a rough score estimate $s(z) = \nabla_z \log p(z)$; (ii) larger apparent per-frame displacements in a tiny system (RMSD $\sim$0.9 Å) making the lDDT threshold stringent; and (iii) step sizes were calibrated to match the reference MD's RMSD-per-frame. For the fast-moving alanine system, this protocol necessitated a large integration step ($\Delta t = 10^{-8}$), which proved numerically aggressive for the Langevin integrator compared to the conservative step ($\Delta t = 10^{-10}$) derived for the slower $A_1AR$ system, pushing the SDE off-manifold.

### 4.3 FIDELITY TO THE EQUILIBRIUM ENSEMBLE

We quantify ensemble fidelity by comparing free–energy surfaces (FES) of backbone $(\phi, \psi)$ and side–chain $(\chi_1, \chi_2)$ dihedrals (Fig. 3). The Jensen–Shannon divergence (JSD) between the normalized model and reference densities serves as our primary error metric (lower is better), with all results summarized in Table 2.

Table 2: **Dihedral free-energy fidelity (JSD; lower is better).** Values represent the mean $\pm$ standard deviation over 5 independent runs. **Bold** indicates the best mean performance.

| Propagator | ADP $(\phi, \psi)$ | $A_1AR$ $(\phi, \psi)$ | $A_1AR$ $(\chi_1, \chi_2)$ |
|---|---|---|---|
| Koopman | $0.138 \pm 0.015$ | $0.043 \pm 0.006$ | $0.121 \pm 0.012$ |
| Autoreg. NN | $\mathbf{0.061 \pm 0.003}$ | $\mathbf{0.019 \pm 0.001}$ | $0.067 \pm 0.002$ |
| Langevin | $0.164 \pm 0.021$ | $0.052 \pm 0.008$ | $\mathbf{0.058 \pm 0.009}$ |

**Backbone $(\phi, \psi)$ fidelity is dominated by the Autoregressive NN.** On backbone dihedrals, the Autoregressive NN consistently outperforms the other propagators in both accuracy and stability. For alanine dipeptide, the NN achieves the lowest divergence (JSD $= 0.061 \pm 0.003$), accurately capturing the canonical Ramachandran basins with high reproducibility. In contrast, the Koopman model's rigidity yields a less accurate landscape (JSD $= 0.138$) with noticeable variance across runs ($\pm 0.015$). This trend is even more pronounced on $A_1AR$, where the NN produces the most faithful backbone surface (JSD $= 0.019$) and negligible variance ($\pm 0.001$), confirming it as the most robust estimator for global backbone topology.

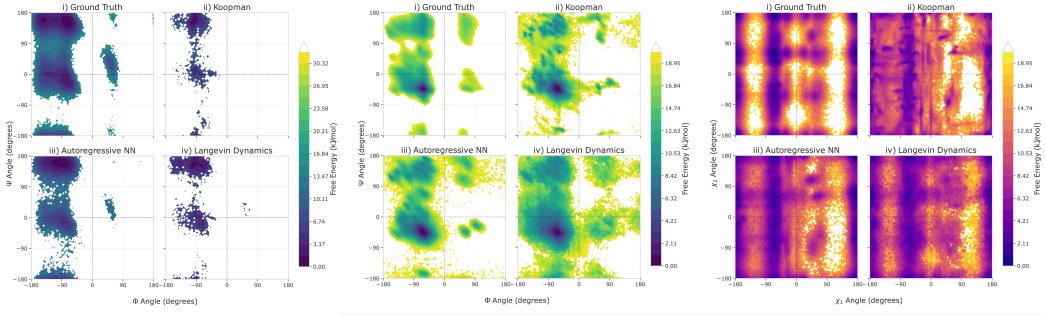

Figure 3: **Ensemble fidelity in dihedral space.** Free-energy maps for backbone $(\phi, \psi)$ and, for $A_1AR$, side-chain $(\chi_1, \chi_2)$. Alanine dipeptide (left) shows canonical basins recovered by all methods; the Autoregressive NN aligns best with the reference basin shapes. For $A_1AR$(middle/right), the NN most closely matches backbone structure, while the score-guided Langevin propagator recovers the sharpest rotamer bands for side-chains.

**Side-chain $(\chi_1, \chi_2)$ rotamers favor the Langevin propagator.** The picture reverses for side-chain rotamers on $A_1AR$. Here, the score-guided *Langevin* propagator achieves the best mean performance (JSD = 0.058), yielding the sharpest rotamer bands. This likely reflects its thermodynamically consistent drift $(T\,s(z))$ plus isotropic noise, which excels at sampling sharp, multi-modal distributions. However, this comes at the cost of higher variance $(\pm 0.009)$ compared to the NN $(\pm 0.002)$, as the stochastic nature of the integrator leads to greater fluctuation between independent trajectories. The NN remains competitive (JSD = 0.067) and highly stable, but slightly oversmooths high-barrier regions compared to the Langevin baseline.

Taken together, these results highlight a three-way trade-off: the **Autoregressive NN** is the most stable and accurate for broad backbone dynamics; the **Score-Guided Langevin** propagator is superior for resolving fine-grained, multi-modal side-chain states albeit with higher stochastic variance; and **Koopman** provides a lightweight but more rigid baseline with moderate variability.

## 4.4 GPCR FUNCTION: ACTIVATION SURFACE IN $A_2AR$

To assess if each propagator reproduces the inactive↔active switching of $A_2AR$, we project its dynamics onto a free-energy surface defined by the TM3–6 and TM3–7 distances (Fig. 4). These coordinates track the outward swing of TM6 and the opening of the intracellular cavity, which are hallmarks of GPCR activation (see App. Fig. 5 for representative snapshots).

**$A_2AR$: two-dimensional surface.** The reference surface forms a diagonal valley, indicating coordinated changes in TM3–6 and TM3–7 during activation. *Langevin* covers this valley most extensively, including the transition corridor connecting inactive and active-like regions. The *Autoregressive NN* follows the same valley with a tighter footprint, giving good alignment near the basin center but under-sampling the flanks. *Koopman* identifies the central basin yet shows stiffer, more isotropic contours; barriers appear sharper and the diagonal anisotropy is reduced, which limits coverage along the transition path.

**One-dimensional slices.** The 1D free energy profiles (Fig. 4, right column) quantify these deviations. Along TM3–7 (top-right), *Langevin* and the *NN* track the position of the principal minimum and its curvature, with *Langevin* retaining broader support across the valley shoulders. *Koopman* exhibits a narrow well and steeper rise away from the minimum, consistent with its tendency to compress variance on the 2D map. Along TM3–6 (bottom-right), *Langevin* again follows the reference profile across both flanks, while the *NN* stays close near the minimum but with a slightly skewed tail. *Koopman* produces a compact basin and elevated shoulders, which reduces sampling of the transition corridor toward larger TM3–6 distances limiting its exploration of the activation corridor and showing that the non-linear propagators are more effective at tracing the complete activation pathway.

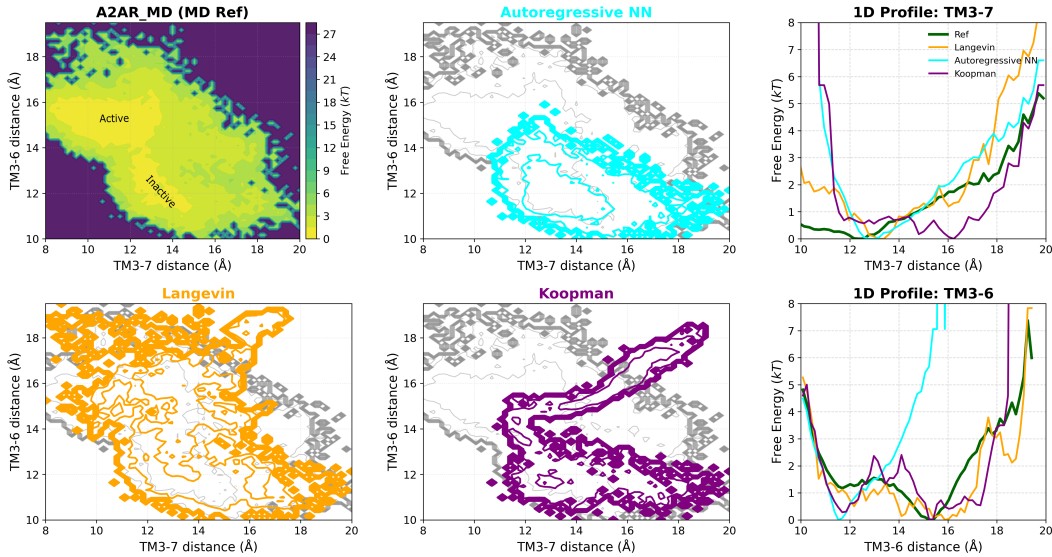

Figure 4: **Functional free–energy surface for A₂AR. (Top Left)** Reference MD free energy heatmap projected onto TM3–6 and TM3–7 distances, showing the transition valley. **(Center & Bottom-Left)** Generated ensembles for the Autoregressive NN (cyan), Langevin (orange), and Koopman (purple) propagators, plotted as contours over the reference background (grey). **(Right Column)** 1D free energy profiles projected along the TM3–7 (top) and TM3–6 (bottom) reaction coordinates.

## 5 OUTLOOK

Our comparison reveals a fundamental trade-off between the stability of data-driven models and the fine-grained physical accuracy of score-guided methods. This points toward several promising future directions. First, motivated by the complementary strengths observed in Table 2 (autoregressive NN for backbone, Langevin for side-chains), we envision hybrid propagators that leverage neural networks for the stable, long-horizon integration of slow collective motions, while employing a score-guided Langevin update for local, high-frequency sampling to ensure thermodynamic correctness. Second, while our TICA analysis confirms that GLDP captures physical timescales and memory effects on complex systems where baselines fail, future work could enforce explicit kinetic constraints during training, such as spectral matching losses, to further refine this fidelity. Third, addressing the need for transferability, moving beyond a fixed backbone to the end-to-end training of the encoder and propagator on multi-protein datasets could discover latent representations optimized for generalizable temporal evolution, potentially bridging the gap between generalist foundation models and the high thermodynamic fidelity required for specific drug targets.

REPRODUCIBILITY STATEMENT.

We release a repository with all source code for the encoder–propagator–decoder pipeline, training, and evaluation, plus fixed random seeds and an environment file to pin dependencies. The main paper outlines the pipeline (Section 3) and the evaluation metrics (Section 4.3). Full implementation details appear in the Appendix: datasets and preprocessing (Appendix A.1), backbone configuration (Appendix A.2), propagators and hyperparameters (Appendix A.3, A.4, A.5), and ablations (Appendix A.8). The repository hosts the preprocessed latent trajectories used in our runs and links to the raw MD sources. To aid inspection, representative structure snapshots for ADP, 7JFL, A₁AR, and A₂AR are included in the Appendix (Figure 5), while short videos of the rollouts are available at the anonymous repository link.

## ACKNOWLEDGMENTS

A.S. was supported by a grant from the Center of Intelligence Systems (EPFL) to P.B. and P.V. This work was also supported by the Institute of Bioengineering (EPFL), the Signal Processing Laboratory (LTS2, EPFL), the Ludwig Institute for Cancer Research, and the Swiss National Science Foundation (grants 31003A_182263 and 310030_208179).

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

# A    APPENDIX

This Supplementary Information provides detailed descriptions of the models, training procedures, and evaluation metrics used in this work to ensure reproducibility.

## A.1    SIMULATION DATASETS AND CODE AVAILABILITY

The complete code, containing the pipeline to train the LD-FPG model and run GLDP, is publicly available at: `https://github.com/adityasengar/GLDP`. The latent trajectories used to train the propagators were derived from three publicly available Molecular Dynamics (MD) simulation datasets. For each system, the original source provided a structure file (PDB) and a coordinate trajectory (XTC). These files were then processed using the LD-FPG framework's preprocessing scripts to generate the inputs for GLDP, including a `condensed.json` file that provides consistent, zero-based atom indexing and defines the atom quadruplets required for calculating all backbone and side-chain dihedral angles.

**Alanine Dipeptide.**    This dataset features N-acetyl-L-alanine-N'-methylamide, a 22-atom molecule commonly known as alanine dipeptide. It is a canonical benchmark for developing and testing new simulation methods due to its simple yet non-trivial conformational landscape, which is primarily described by its two backbone dihedral angles $(\phi, \psi)$. The data was sourced from the CMB data repository at `ftp.imp.fu-berlin.de` and consists of a 250 ns simulation trajectory with solvent molecules removed.

**7JFL    (ATLAS).**    We use entry `7jfl_C` from the ATLAS database: `dsimb.inserm.fr/ATLAS/.../7jfl_C`. ATLAS provides standardized all-atom, explicit-solvent MD trajectories across proteins under a common protocol.

**Adenosine $A_1$ receptor ($A_1AR$).**    Human $A_1$ARtrajectories were taken from the study  (D'Amore et al., 2024). These are explicit-solvent, membrane-embedded GPCR simulations as described in the paper's Methods/SI and associated data records. Before encoding, we remove solvent and lipids and operate at fixed box; we assess configuration-space statistics after decoding. Where specific thermostat/barostat and T/P values are required, we defer to the original Methods/SI.

**HIV-1 Protease (1R6W).**    To evaluate performance on mixed secondary structures ($\alpha$-helices and $\beta$-sheets), we utilized the HIV-1 Protease system (PDB: 1R6W). We sourced the MD trajectory from the ATLAS databse `dsimb.inserm.fr/ATLAS/.../1r6W`.

**Adenosine $A_{2A}$ receptor ($A_2AR$).**    $A_2$ARtrajectories come from the  (D'Amore et al., 2024). As above, simulations are explicit solvent with a membrane environment; solvent and lipids are stripped before encoding. We match configuration-space statistics to the reference ensemble and refer to the paper's Methods/SI for the exact simulation state (e.g., NVT/NPT, temperature, pressure, ions, water model, force field).

## A.2    LD-FPG BASED ENCODER-DECODER BACKBONE

GLDP follows an encoder-propagator-decoder blueprint. We use a fixed, pre-trained encoder-decoder from the LD-FPG model (Sengar et al., 2025) and focus on comparing propagators within its learned latent space.

The latent space for this study was generated using the original LD-FPG framework. The core components are:

- **Encoder**: A 4-layer Chebyshev Graph Neural Network ('HNO' model) that maps all-atom coordinates $X(t) \in \mathbb{R}^{N \times 3}$ to per-atom embeddings $Z(t) \in \mathbb{R}^{N \times d_z}$.
- **Pooling and Decoder**: A 'ProteinStateReconstructor2D' model performs pooling on the per-atom embeddings to generate a single latent vector $z(t) \in \mathbb{R}^{d_{\text{flat}}}$ for the entire conformation. The decoder portion of this model maps this latent vector back to all-atom coordinates.

The script 'chebet_blind.py' was used to train this autoencoder and export the time-series of latent vectors $\{z_0, z_1, \ldots, z_M\}$ into an HDF5 file ('latent_reps/pooled_embedding.h5'), which serves as the ground-truth trajectory for training the propagators. Key hyperparameters for the backbone are detailed in Table 3.

Table 3: Configuration of the Pre-trained LD-FPG Backbone Model. *Note: Pooling is performed immediately after the graph convolutions to form the latent space.*

| Component | Parameter Value |
|---|---|
| **Encoder & Pooling (Latent Construction)** | |
| Graph Type | Chebyshev GNN (HNO) |
| Hidden Dimension ($d_z$) | 16 |
| Chebyshev Order ($K$) | 4 |
| Graph Construction | $k$-Nearest Neighbors ($k = 4$) |
| Pooling Type | Blind |
| Pooling Output Size (Latent Bottleneck) | 100 |
| **Decoder (Reconstruction)** | |
| MLP Hidden Layers | 12 |
| MLP Hidden Dimension | 128 |

**Encoder (HNO).**  HNO denotes a Chebyshev graph neural operator: a ChebNet with polynomial filters of order $K$ on the molecular graph (built by $k$NN).

**Pooling.**  "Blind pooling" maps per–atom embeddings $Z(t) \in \mathbb{R}^{N \times d_z}$ to a fixed $H \times W$ latent by a learned MLP that aggregates across atoms without explicit residue topology; the decoder inverts this map to all-atom coordinates.

### A.3  KOOPMAN OPERATOR IMPLEMENTATION DETAILS

The Koopman operator $A$ was computed using the `fit_dmd` function, which implements the standard Dynamic Mode Decomposition (DMD) algorithm. The procedure is summarized in Algorithm 1. This method provides a numerically robust way to find the best-fit linear operator $A$ by leveraging the Singular Value Decomposition (SVD) to compute the pseudoinverse of the snapshot matrix $\mathbf{X}$.

---

**Algorithm 1** Computing the Koopman operator with (truncated) DMD

---

**Require:** Latent snapshots $\{z_t\}_{t=0}^{M-1}$, rank cap $r$ (optional), tolerance $\varepsilon$ (optional).
 1: Construct $X = [z_0, \ldots, z_{M-2}], Y = [z_1, \ldots, z_{M-1}]$.
 2: Compute SVD: $X = U\Sigma V^{\top}$.
 3: (Optional) choose rank $r$ by energy or cap; set $U_r, \Sigma_r, V_r$ to the top-$r$ factors.
 4: Form $\Sigma_r^{+}$ by inverting only singular values $> \varepsilon$ (set others to zero).
 5: Compute $X^{+} \leftarrow V_r \Sigma_r^{+} U_r^{\top}$.
 6: Set $A \leftarrow Y X^{+}$.
**Ensure:** Linear propagator $A$ (use $z_{t+1} = Az_t$, optionally with small Gaussian rollout noise).

---

In our implementation, the SVD truncation rank $r$ can be set as a hyperparameter to de-noise the dynamics. If no rank is specified, the full rank of the matrix is used.

### A.3.1  KOOPMAN PROPAGATOR HYPERPARAMETERS

The SVD truncation rank ($r$) and the standard deviation of the Gaussian rollout noise ($\sigma_{\text{koop}}$) are key hyperparameters. They were selected independently for each system to ensure stable, long-horizon rollouts, with the final values reported in Table 4.

Table 4: Koopman model hyperparameters used for each molecular system.

| System | SVD Truncation Rank ($r$) | Rollout Noise Std. ($\sigma_{\textbf{koop}}$) |
|---|---|---|
| $A_1AR$ | 10 | $2.00 \times 10^{-4}$ |
| Alanine Dipeptide | 10 | $2.60 \times 10^{-3}$ |

## A.4  AUTOREGRESSIVE NEURAL NETWORK DETAILS

The neural network propagator $f_\theta$ is implemented as a Multi-Layer Perceptron using PyTorch, as detailed in the `neural_propagator.py` script. The training and rollout procedures are outlined in Algorithm 2, and the specific hyperparameters are summarized in Table 5. The model is trained on a fixed percentage of the latent trajectory, with the remainder used for validation to select the best-performing model checkpoint.

Table 5: Hyperparameters for the Autoregressive NN Propagator.

| Parameter | Value / Description |
|---|---|
| **Architecture** | |
| Model Type | MLP (`PropagatorMLP`) |
| Input Dimension ($d_{\text{flat}}$) | 100 |
| Hidden Dimension | 512 |
| Hidden Layers | 2 |
| Activation Function | ReLU |
| Dropout Rate | 0.1 |
| **Training** | |
| Loss Function | Mean Squared Error (MSE) |
| Optimizer | Adam |
| Learning Rate | $1 \times 10^{-4}$ |
| Batch Size | 128 |
| Epochs | 400 |
| Frame Skip ($n$ in $z_{t+n}$) | 1 |
| Training Fraction | 0.8 (80% train, 20% validation) |
| Weight Decay | $1 \times 10^{-4}$ |

---

**Algorithm 2** Autoregressive NN Training and Rollout

---

**Require:** Latent trajectory $\{z_0, \ldots, z_M\}$, model $f_\theta$, learning rate $\alpha$.
1:  **// Training**
2:  Create dataset of pairs $\mathcal{D} = \{(z_t, z_{t+1})\}_{t=0}^{M-2}$.
3:  Initialize model parameters $\theta$.
4:  **for** each epoch **do**
5:      **for** each batch $(z_t, z_{t+1})$ in $\mathcal{D}$ **do**
6:          Compute loss: $\mathcal{L} \leftarrow \|z_{t+1} - f_\theta(z_t)\|^2$.
7:          Update parameters via backpropagation: $\theta \leftarrow \theta - \alpha \nabla_\theta \mathcal{L}$.
8:      **end for**
9:  **end for**
10:
11: **// Rollout**
12: Initialize trajectory: $\hat{z}_0 \leftarrow z_0$.
13: **for** $t = 0, \ldots, M-2$ **do**
14:     $\hat{z}_{t+1} = f_\theta(\hat{z}_t) + \eta_t, \quad \eta_t \sim \mathcal{N}(0, \sigma_{\text{roll}}^2 I)$.
15: **end for**
**Ensure:** Trained model $f_\theta$, generated trajectory $\{\hat{z}_t\}$.

---

We choose $\sigma_{\text{roll}}$ on a validation split to match the one-step latent variance of $z_{t+1} - z_t$.

The standard deviation of the rollout noise ($\sigma_{\text{roll}}$) was calibrated for each system by matching the one-step latent residual variance on a validation set, as described in the main text. The specific values used for each simulation are listed in Table 6.

Table 6: System-specific rollout noise for the Autoregressive NN propagator.

| System | Rollout Noise Std. ($\sigma_{\text{roll}}$) |
|---|---|
| $A_1AR$ | $1.00 \times 10^{-3}$ |
| Alanine Dipeptide | $1.00 \times 10^{-2}$ |
| 7JFL | $1.50 \times 10^{-3}$ |

## A.5   SCORE-GUIDED LANGEVIN PROPAGATOR DETAILS

The implementation of the Score-Guided Langevin Propagator consists of two main stages.

### A.5.1   DENOISER MODEL TRAINING

First, a time-conditional denoising model, $\epsilon_\theta(z_\tau, \tau)$, is trained on the static set of latent embeddings from the MD trajectory. This model learns to predict the noise that was added to a "clean" sample $z_0$ to produce a noisy sample $z_\tau$ at diffusion step $\tau$. The training objective is to minimize the Mean Squared Error between the predicted noise and the true noise: $\mathcal{L} = \mathbb{E}_{t,z_0,\epsilon}\|\epsilon - \epsilon_\theta(\sqrt{\bar{\alpha}_\tau}z_0 + \sqrt{1 - \bar{\alpha}_\tau}\epsilon, \tau)\|^2$. The key hyperparameters for this training stage are listed in Table 7.

Table 7: Hyperparameters for Denoiser Model Training.

| Parameter | Value / Description |
|---|---|
| **Denoiser Model** | |
| Model Type | Time-conditional MLP (`DiffusionMLP_v2`) |
| Input Dimension ($d_{\text{flat}}$) | 100 |
| Hidden Dimension | 1024 |
| **Diffusion Schedule** | |
| Diffusion Steps ($T_{\text{diff}}$) | Range explored: $450 - 1600$ |
| Beta Schedule | Linear |
| $\beta_{\text{start}}$ | Range explored: $5 \times 10^{-6} - 0.005$ |
| $\beta_{\text{end}}$ | Range explored: $0.02 - 0.11$ |
| **Training** | |
| Optimizer | Adam |
| Learning Rate | $1 \times 10^{-5}$ |
| Batch Size | 64 |
| Epochs | 50,000 |

### A.5.2   LANGEVIN SIMULATION

We approximate the score of the *unperturbed* data distribution by evaluating the model at a low, fixed noise level, $\tau_{\text{noise}}$ (e.g., $\tau = 0$). The simulation then proceeds according to Algorithm 3, with simulation-specific hyperparameters listed in Table 8.

Table 8: Hyperparameters for Langevin Simulation.

| Parameter | Value / Description |
|---|---|
| Integration Time Step ($\Delta t$) | Typical range: $10^{-11} - 10^{-8}$ (arbitrary units) |
| Temperature ($T$) | Typical range: $0.001 - 1.0$ (arbitrary units) |
| Noise Level Timestep ($\tau_{\text{noise}}$) | 0 (Evaluates score at lowest noise) |
| Score Clipping Norm | Optional max norm to prevent instability |

---

**Algorithm 3** Score-Guided Langevin Simulation

**Require:** Pre-trained denoiser $\epsilon_\theta$, initial state $z_0$, time step $\Delta t$, temperature $T$, simulation steps $N_{steps}$.
 1: Fix a low noise level for score evaluation: $\tau \leftarrow \tau_{\text{noise}}$.
 2: Look up noise standard deviation for $\tau$: $\sigma_\tau$.
 3: Initialize trajectory: $\mathcal{Z} \leftarrow \{z_0\}$.
 4: **for** $t = 0, \ldots, N_{steps} - 1$ **do**
 5:    **// 1. Compute Score**
 6:    Predict noise for the current state $z_t$: $\epsilon_{pred} \leftarrow \epsilon_\theta(z_t, \tau)$.
 7:    Calculate the score: $s(z_t) \leftarrow -\epsilon_{pred}/\sigma_\tau$.
 8:    (Optional) Clip score: if $\|s(z_t)\| > s_{max}$, then $s(z_t) \leftarrow s(z_t) \cdot s_{max}/\|s(z_t)\|$.
 9:
10:    **// 2. Update State**
11:    Calculate drift: $\Delta z_{\text{drift}} \leftarrow T\Delta t \cdot s(z_t)$.
12:    Calculate diffusion: $\Delta z_{\text{diff}} \leftarrow \sqrt{2T\Delta t} \cdot \mathcal{N}(0, I)$.
13:    Update state: $z_{t+1} \leftarrow z_t + \Delta z_{\text{drift}} + \Delta z_{\text{diff}}$.
14:    Append $z_{t+1}$ to $\mathcal{Z}$.
15: **end for**
**Ensure:** Generated trajectory $\mathcal{Z}$.

---

As noted in the main text, key parameters such as the temperature ($T$), time step ($\Delta t$), and sampling stride ($s$) were calibrated for each system to ensure physically realistic and stable trajectories. The specific values used for each protein simulation are provided in Table 9.

Table 9: System-specific simulation parameters for the Score-Guided Langevin propagator. Temperature ($T$) and time step ($\Delta t$) are in latent units.

| System | Temperature ($T$) | Time Step ($\Delta t$) | Total SDE Steps | Sampling Stride ($s$) |
|---|---|---|---|---|
| Alanine Dipeptide | 5.0 | $1.0 \times 10^{-8}$ | 10,000 | 1 |
| $A_1AR$ | 0.8 | $1.0 \times 10^{-10}$ | 100,000 | 20 |
| $A_2AR$ | 0.8 | $1.0 \times 10^{-10}$ | 100,000 | 20 |

**Implementation and Calibration Details.** At each step of the Langevin simulation, we evaluate the score $s(z_t)$ by calling the denoiser $\epsilon_\theta(z_t, t_{\text{noise}})$ and dividing by $\sqrt{1 - \bar{\alpha}_{t_{\text{noise}}}}$. To prevent numerical instability from rare large updates, we clip the score's L2-norm to a maximum value of $c = 10$. We optionally subsample the simulated path by emitting one decoded frame every $s$ SDE steps (sampling stride) to control the apparent per-frame displacement. The temperature $T$ and step size $\Delta t$ are critical hyperparameters, which we calibrate for each system to match the short-horizon decoded RMSD-per-frame of the reference MD trajectory (as described in Section 4.2). Additional training and rollout settings, including the diffusion schedule and optimizer, are also detailed in this section.

A.6   CONTACT MAP PEARSON CORRELATION

Given a topology $P$ and a trajectory $X = \{x_t\}_{t=1}^T$, we build a C$\alpha$–based contact map as follows. For each frame $t$, extract C$\alpha$ coordinates and compute the pairwise Euclidean distance matrix $D^{(t)}$ (all

residues). Average over frames to obtain $\bar{D} = \frac{1}{T} \sum_{t=1}^{T} D^{(t)}$, then threshold at a fixed cutoff $\delta$ (8 Å by default) to form a *binary* contact map $C \in \{0, 1\}^{N \times N}$:

$$C_{ij} = \mathbf{1}\big[\bar{D}_{ij} < \delta\big], \qquad C_{ii} = 0 \ \forall i.$$

We construct $C^{\mathrm{ref}}$ and $C^{\mathrm{mod}}$ with the *same* cutoff and residue selection. The reported score is the Pearson correlation coefficient between the flattened matrices:

$$r = \mathrm{corr}\big(\mathrm{vec}\big(C^{\mathrm{ref}}\big), \mathrm{vec}\big(C^{\mathrm{mod}}\big)\big),$$

*Implementation notes.* The procedure uses MDAnalysis for I/O and atom selection (`name CA`); distances are computed via `pdist` and `squareform`, then averaged over frames prior to thresholding. No exclusion of near-sequence pairs ($|i - j| < 3$) is applied in this metric.

## A.7 METHODOLOGY FOR KINETIC COMPARISON

To quantitatively evaluate the kinetic properties of the generated molecular dynamics trajectories against the ground truth simulations, we employed a two-stage analysis based on Time-lagged Independent Component Analysis (TICA) and Autocorrelation Functions (ACF). This approach allows us to identify the slowest collective motions within the system and measure their characteristic timescales.

### A.7.1 STEP 1: FEATURIZATION

The high-dimensional trajectory data was first transformed into a feature space suitable for kinetic analysis. We used the pairwise distances between all C-alpha atoms as our features. This choice of internal coordinates ensures that the analysis is invariant to rotational and translational motions of the protein as a whole. For a protein with $N$ C-alpha atoms, this results in a feature vector $x(t) \in \mathbb{R}^{N(N-1)/2}$ for each time frame $t$.

### A.7.2 STEP 2: TIME-LAGGED INDEPENDENT COMPONENT ANALYSIS (TICA)

TICA is a powerful dimensionality reduction technique that finds the slowest, most dynamically relevant degrees of freedom from a molecular dynamics simulation. The objective of TICA is to find a linear combination of input features that maximizes the autocorrelation of the resulting components at a specified lag time, $\tau$. This is achieved by solving the generalized eigenvalue problem:

$$C(\tau)v_i = \lambda_i C(0)v_i \tag{6}$$

where:

- $C(0)$ is the standard covariance matrix of the feature vector $x(t)$:

$$C(0) = \mathbb{E}\left[(x(t) - \mu)(x(t) - \mu)^T\right] \tag{7}$$

- $C(\tau)$ is the time-lagged covariance matrix at a lag time $\tau$:

$$C(\tau) = \mathbb{E}\left[(x(t) - \mu)(x(t + \tau) - \mu)^T\right] \tag{8}$$

- $\mu = \mathbb{E}[x(t)]$ is the mean of the feature vector.

- $\lambda_i$ are the eigenvalues, which represent the autocorrelation of the $i$-th component at lag time $\tau$.

- $v_i$ are the eigenvectors, which are the TICA components (TICs) or collective coordinates.

The TICA model was fit exclusively on the ground truth trajectory data to define the principal slow kinetic modes of the system.

### A.7.3   STEP 3: CALCULATION OF IMPLIED TIMESCALES

The eigenvalues $\lambda_i$ obtained from the TICA decomposition are directly related to the characteristic "implied timescale" $t_i$ of each component. This timescale represents the decorrelation time of the corresponding TIC. The relationship is given by:

$$t_i = -\frac{\tau}{\ln|\lambda_i|} \tag{9}$$

The largest eigenvalue, $\lambda_1$, corresponds to the slowest process in the system, and its associated implied timescale, $t_1$, provides a quantitative measure of the duration of this dominant kinetic event. We report this slowest timescale as a key metric for kinetic validity.

### A.7.4   STEP 4: PROJECTION ONTO THE SLOWEST MODE

Once the TICA model was trained on the ground truth data, both the ground truth and the generated trajectories were projected onto the first (slowest) independent component, TIC1 (corresponding to eigenvector $v_1$). This projection reduces the high-dimensional conformational dynamics to a one-dimensional time series, $z(t)$, that captures the system's evolution along its most significant slow coordinate.

$$z(t) = v_1^T(x(t) - \mu) \tag{10}$$

### A.7.5   STEP 5: AUTOCORRELATION AND DECORRELATION TIME

To provide a more direct comparison of the system's "memory" along the slowest mode, we computed the normalized autocorrelation function (ACF) of the time series $z(t)$ for both trajectories. The ACF at a time lag $\Delta t$ is defined as:

$$A(\Delta t) = \frac{\mathbb{E}\left[(z(t) - \bar{z})(z(t + \Delta t) - \bar{z})\right]}{\sigma_z^2} \tag{11}$$

where $\bar{z}$ is the mean and $\sigma_z^2$ is the variance of the time series $z(t)$. The decorrelation time, $t_{decorr}$, is formally defined as the time lag at which the ACF decays to $1/e$.

$$A(t_{decorr}) = \frac{1}{e} \approx 0.368 \tag{12}$$

This value quantifies how long it takes for the system's conformation along TIC1 to become statistically independent of its initial state. A close match in the decorrelation times between the ground truth and the generated trajectory indicates high kinetic fidelity.

### A.8   ABLATION STUDIES ON $A_1AR$

We probe three levers that materially affect $A_1AR$: (i) encoder per-atom width $d_z$, (ii) pooled latent size prior to decoding (with fixed $W{=}2$ so $d_{\text{flat}}{=}2H$), and (iii) Langevin/denoiser settings. We report (i) training-time losses at the level where the lever acts, and (ii) downstream accuracy via *backbone/side-chain* $\Delta$RMSF (mean absolute error, Å) and *lDDT failure time* (first frame with lDDT$< 0.65$; larger is better). For long-rollout statistics we keep the autoregressive NN and Koopman unchanged and vary only the component under study.

**Conventions.**   For Langevin rollouts, the sampling stride is $s{=}1$ (one SDE step per frame). We choose $\Delta t$ by matching the *average per-frame RMSD* of the decoded trajectory to the MD reference over $K{=}100$ frames (Sec. 4.2). Unless stated: denoiser query $\tau{=}0$, drift clipping $\|s(z)\| \leq c{=}10$.

### A.8.1 (I) ENCODER PER-ATOM WIDTH $d_z \in \{8, 16, 32\}$

Increasing $d_z$ lowers the autoencoder validation MSE as expected, and modestly improves downstream flexibility metrics, with diminishing returns beyond $d_z=16$.

| $d_z$ | Enc. MSE ↓ | $\Delta$RMSF$_{BB}$ (Å) ↓ | $\Delta$RMSF$_{SC}$ (Å) ↓ | lDDT failure (frames)$^\dagger$ |
|---|---|---|---|---|
| 8 | 0.020 | 0.34 | 0.54 | NN $>$10,000 \| Lang 6,800 \| Koop 5,200 |
| 16 | 0.006 | 0.26 | 0.39 | NN $>$10,000 \| Lang 7,400 \| Koop 5,600 |
| 32 | 0.003 | 0.24 | 0.37 | NN $>$10,000 \| Lang 7,400 \| Koop 5,600 |

$^\dagger$ $A_1$AR, criterion lDDT$<$ 0.65; *Lang* uses $s=1$, $\Delta t=2\times10^{-9}$ (calibrated), $\tau=0$, clip $c=10$.

We select $d_z=16$ in the main runs for a good trade-off between training cost and downstream accuracy.

### A.8.2 (II) POOLING WIDTH $H$ WITH $W=2$ (SO $d_{FLAT}=2H$)

Larger pooled latents reduce decoder MSE, but very large $d_{flat}$ slightly degrades flexibility/stability, likely by making the latent stiffer to traverse with simple propagators.

| $H$ ($d_{flat}$) | Dec. MSE ↓ | $\Delta$RMSF$_{BB}$ (Å) ↓ | $\Delta$RMSF$_{SC}$ (Å) ↓ | lDDT fail (Lang) | lDDT fail (Koop) |
|---|---|---|---|---|---|
| 20 (40) | 0.43 | 0.31 | 0.49 | 7,100 | 5,400 |
| **50 (100)** | **0.24** | **0.26** | **0.39** | **7,400** | **5,600** |
| 80 (160) | 0.21 | 0.30 | 0.46 | 7,000 | 5,500 |

We therefore adopt $H=50$ (i.e., $d_{flat}=100$) for all main results.

### A.8.3 (III) DENOISER/LANGEVIN SETTINGS

We ablate the diffusion schedule for the denoiser (which controls score quality), the score query level $\tau$, the clipping threshold, and the SDE step size $\Delta t$ (with stride fixed to 1). Trends:

- **Diffusion schedule.** Linear $(\beta_{start}, \beta_{end}, T_{diff})$ of $(5\times10^{-6}, 0.03, 1400)$ reached denoiser MSE 0.015 (best); a looser schedule $(5\times10^{-5}, 0.11, 450)$ gave 0.050 (worst). Better denoisers reduce $\Delta$RMSF$_{SC}$ by $\approx 0.03$ Å.

- **Score query level.** $\tau=0$ outperformed a small nonzero level ($\tau=20$) on side-chains: $\Delta$RMSF$_{SC}=0.39$ Å$(\tau=0)$ vs 0.42 Å$(\tau=20)$; backbone was unchanged within 0.01 Å.

- **Clipping.** Max-norm clip at $c=10$ stabilized occasional spikes without biasing small updates; removing clip reduced the Langevin failure time from 7,400 to 6,600 frames; $c=5$ was slightly over-damped (failure 7,200).

- **Step size.** With $s=1$, selecting $\Delta t$ by RMSD matching (Sec. 4.2) gave $\Delta t=2\times10^{-9}$ for $A_1$AR. Varying $\Delta t$ by $\times 2$ changed $\Delta$RMSF values by $\leq 0.02$ Å but affected failure time ($\pm$200–300 frames).

**Summary.** Bigger $d_z$ and pooled width reduce reconstruction losses as expected, but overly large pooled latents slightly worsen flexibility and stability. A well-trained denoiser with $\tau=0$, clip $c=10$, and $\Delta t$ calibrated by *per-frame RMSD matching* yields the best side-chain fidelity and stable long rollouts on $A_1$AR.

### A.9 ABLATION STUDIES ON DATA EFFICIENCY AND TEMPORAL RESOLUTION

To assess the robustness of the Graph Latent Dynamics Propagator (GLDP) beyond the standard benchmark settings, we performed targeted ablations on the **7JFL** system. Specifically, we investigate the dependence of the Autoregressive NN propagator on training data volume, the impact of increasing the temporal prediction stride (frame skipping), and the benefit of residual parameterization.

A.9.1 Dependence on Trajectory Length (Data Efficiency)

We trained the Autoregressive NN propagator on subsets of the full 7JFL MD trajectory, ranging from 5% to 100% of the available frames. We evaluated structural fidelity via Backbone Dihedral JSD and kinetic fidelity via the slowest TICA timescale ($t_1$). We also report the Validation MSE, defined as the mean squared error between the predicted latent state $\hat{z}_{t+1}$ and the ground truth $z_{t+1}$ on a held-out test set, to quantify the local predictive accuracy of the propagator.

As shown in Table 10, the model exhibits a sharp performance threshold. At extremely low data regimes (5%), the propagator fails to learn the underlying physics, resulting in a collapse of the slow dynamics ($t_1 \approx 22$) and poor structural recovery. While structural equilibrium (JSD) begins to stabilize around 25% data usage.

Table 10: **Data Efficiency on 7JFL (Autoregressive NN).** Structural and kinetic metrics as a function of training data fraction. (Mean of 3 runs).

| Data Fraction | Validation MSE ($10^{-4}$) | Backbone JSD ($\downarrow$) | TICA $t_1$ (steps) |
|---|---|---|---|
| 5% | 25.5 | 0.145 | 22.1 |
| 10% | 7.2 | 0.065 | 95.4 |
| 25% | 4.8 | 0.043 | 188.2 |
| 50% | 4.1 | 0.039 | 215.6 |
| 75% | 3.8 | 0.038 | 223.8 |
| **100% (Main Paper)** | **3.7** | **0.037** | **226.4** |

A.9.2 Impact of Frame Skipping (Temporal Stride)

Standard training minimizes the one-step error $z_t \to z_{t+1}$. To test the model's capability to model coarser timescales, we trained separate models to predict $z_t \to z_{t+k}$ for stride $k \in \{1, \ldots, 10\}$.

Table 11 reports the validation MSE and the TICA timescale ($t_1$) in simulation steps. To compare kinetics fairly, we calculate the "Implied Physical Time" ($t_1 \times k$). We observe a systematic reduction in physical memory as the stride increases. For $k = 1$ to $k = 3$, the physical timescale remains robust ($\approx 200+$ units). However, at larger strides ($k = 10$), the implied physical time drops significantly. This indicates that predicting over large temporal gaps may cause the network to "smooth out" energy barriers, leading to accelerated transition kinetics and a loss of long-term memory.

Table 11: **Effect of Frame Skipping on 7JFL.** Validation MSE and Kinetic Fidelity. Note that $t_1$ is reported in simulation steps; Implied Physical Time is $t_1 \times k$.

| Stride ($k$) | Validation MSE ($10^{-4}$) | TICA $t_1$ (steps) | Implied Phys. Time ($t_1 \times k$) |
|---|---|---|---|
| 1 (Baseline) | **3.7** | **226.4** | **226.4** |
| 2 | 3.9 | 108.5 | 217.0 |
| 3 | 4.2 | 68.2 | 204.6 |
| 4 | 4.8 | 46.1 | 184.4 |
| 5 | 6.5 | 30.5 | 152.5 |
| 10 | 15.2 | 8.4 | 84.0 |

A.9.3 Residual vs. Direct Prediction

Finally, we addressed the architectural choice of predicting the next state directly, $z_{t+1} = f_\theta(z_t)$ (as used in the main paper), versus predicting the residual, $z_{t+1} = z_t + f_\theta(z_t)$.

As detailed in Table 12, both architectures converge to nearly identical performance. The residual formulation achieves a marginally lower validation MSE (3.6 vs 3.7), but the resulting physical kinetics (TICA $t_1$) are statistically indistinguishable (228 vs 226). This confirms that while residual learning is a valid optimization, the Direct Prediction formulation used in our main results is sufficient to capture the transition dynamics accurately.

Table 12: **Architectural Comparison on 7JFL.** Validation error and kinetic fidelity.

| Architecture | Validation MSE $(10^{-4})$ | TICA $t_1$ (steps) |
|---|---|---|
| Direct Prediction (Main Paper) | 3.7 | 226.4 |
| Residual Prediction | **3.6** | **228.1** |

### A.9.4 COMPUTATIONAL EFFICIENCY OF LATENT PROPAGATORS

To quantify the utility of each propagator for long-horizon simulations, we benchmarked the wall-clock time required to generate trajectories of varying lengths. This analysis was performed for the **7JFL** protein system on a single NVIDIA **L40S GPU**. The results clarify the trade-off between physical accuracy and raw speed.

Table 13: Wall-Clock Time for Trajectory Generation on 7JFL (L40S GPU)

| Number of Steps | Koopman (Average) | Autoregressive NN | Score-Guided Langevin |
|---|---|---|---|
| 1,000 | 0.15s | 2.62s | 2.17s |
| 10,000 | 0.19s | 3.42s | 4.03s |
| 100,000 | 0.59s | 12.53s | 23.88s |

### A.10 ACKNOWLEDGEMENT OF LLM USAGE

During the preparation of this work, we utilized Large Language Models (LLMs) to assist with literature discovery, information retrieval, and the initial drafting of the manuscript. The authors reviewed, edited, and validated all LLM-assisted text and take full responsibility for the accuracy, originality, and integrity of the final content.

### A.11 SUPPLEMENTARY FIGURES

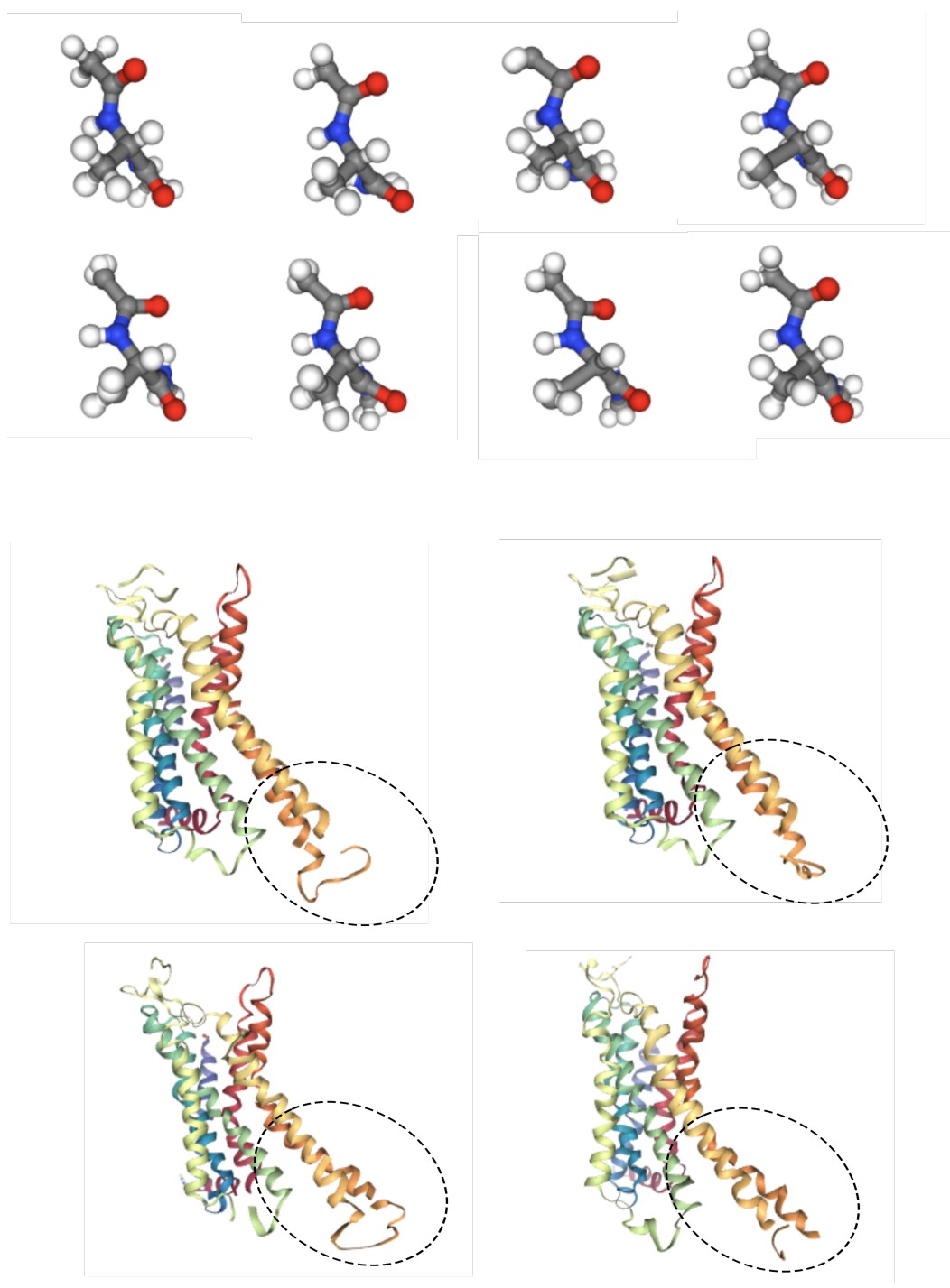

Figure 5: **Representative structural snapshots from latent rollouts.** *Top:* Alanine–dipeptide conformers sampled from a **Koopman** rollout show backbone dihedral changes across frames. *Bottom:* $A_2AR$ snapshots from an **Langevin** rollout; the dashed circle highlights the intracellular end of TM6, which moves outward over time—a hallmark of GPCR activation.

A.12    SUPPLEMENTARY TABLES

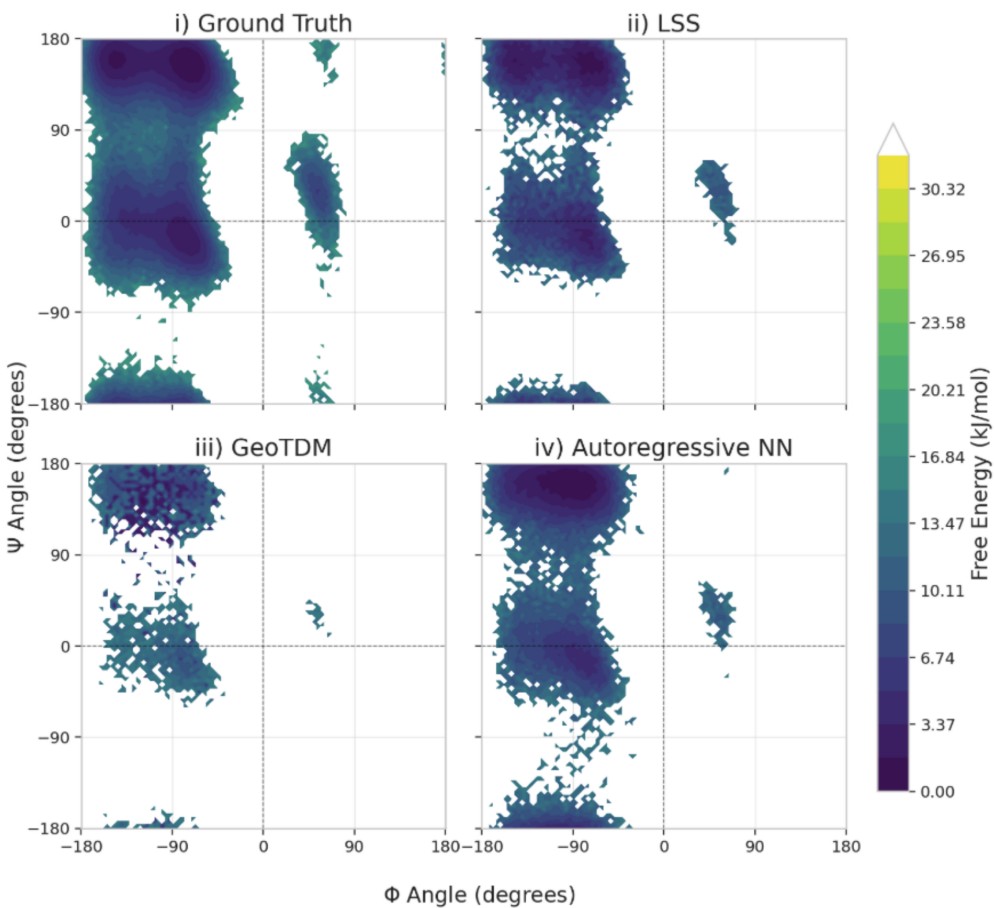

Figure 6: **Alanine Dipeptide Backbone Free Energy.** Comparison of the $(\phi, \psi)$ free energy surface for (i) Ground Truth MD, (ii) LSS, (iii) GeoTDM, and (iv) GLDP (Autoregressive NN).

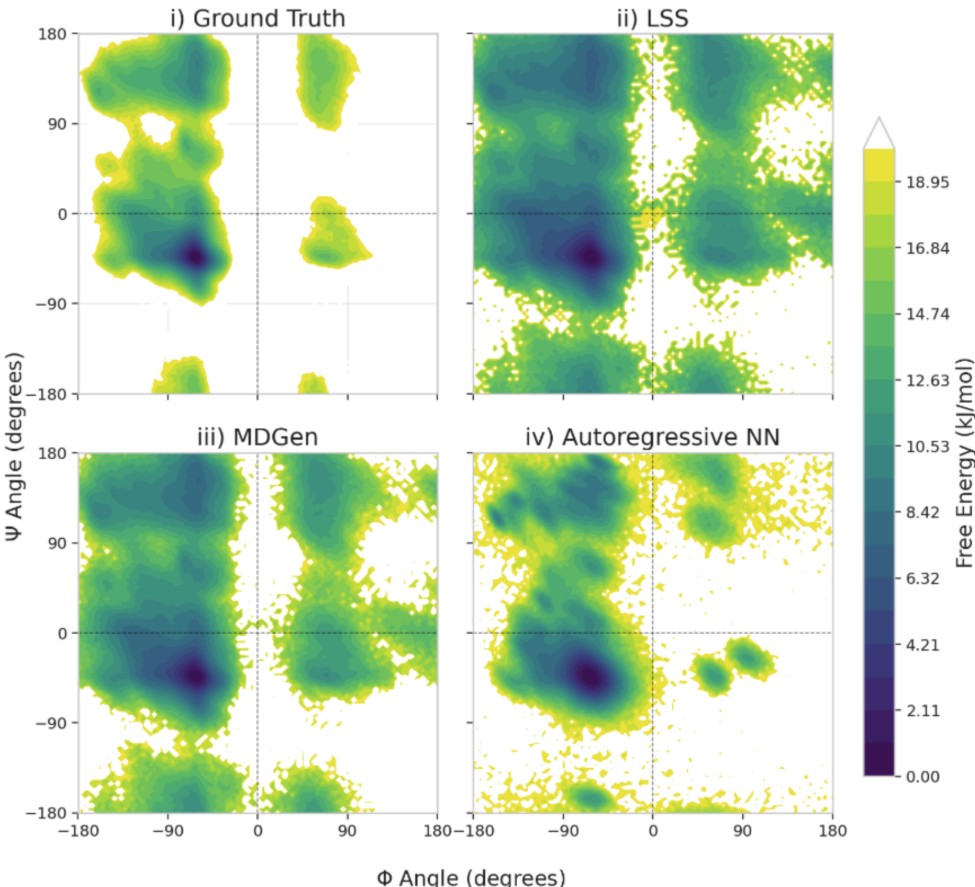

Figure 7: **A$_1$AR Backbone Free Energy.** Comparison of aggregated backbone $(\phi, \psi)$ distributions. LSS (ii) shows a noisy, high-entropy landscape. GLDP (iv) and MD-Gen (iii) successfully recover the tight $\alpha$-helical basins observed in the Ground Truth (i).

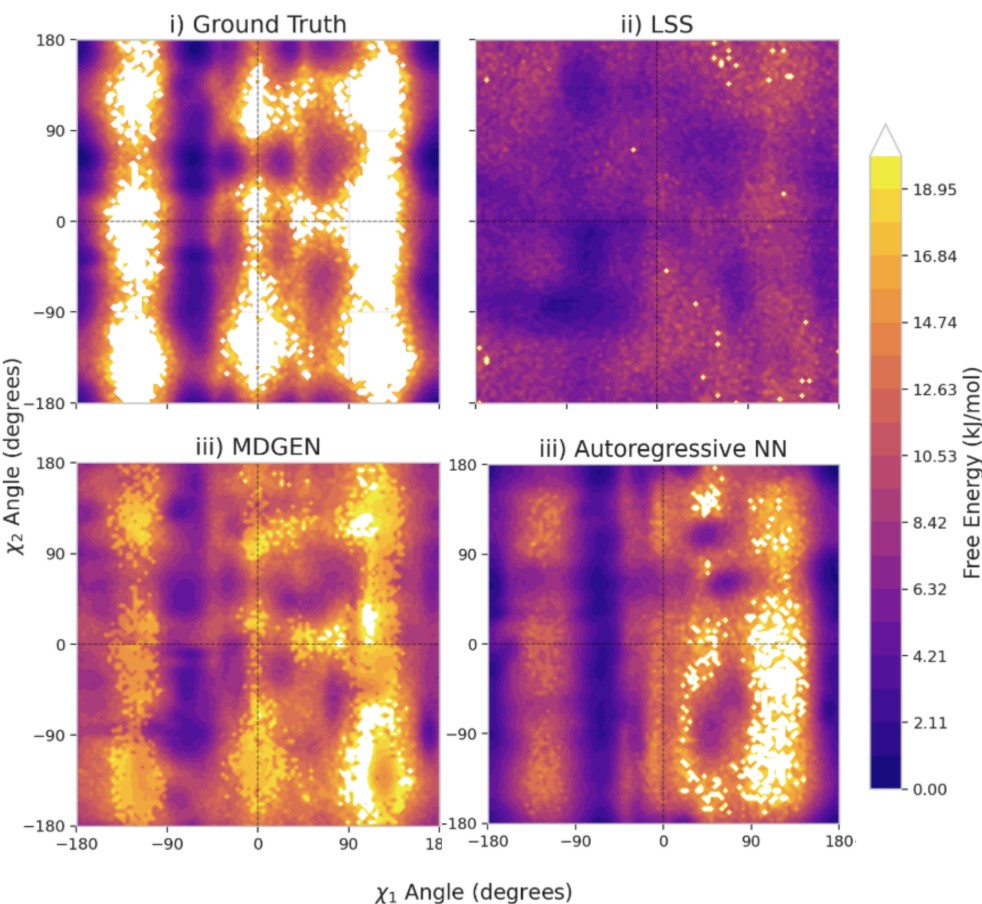

Figure 8: **A₁AR Side-chain Free Energy.** Comparison of aggregated side-chain $(\chi_1, \chi_2)$ distributions. The ability to capture rotameric packing is critical for stability. LSS (ii) fails to resolve clear rotamer wells. GLDP (iv) accurately reproduces the multi-modal rotameric density seen in the Ground Truth (i).

Table 14: Summary of notation used in the description of the propagator models.

| Symbol | Description |
|---|---|
| $z_t \in \mathbb{R}^d$ | Latent space vector at time $t$. |
| $\mathbf{X}, \mathbf{Y} \in \mathbb{R}^{d \times (M-1)}$ | Snapshot matrices with *columns* as time: $\mathbf{X} = [z_0, \ldots, z_{M-2}]$, $\mathbf{Y} = [z_1, \ldots, z_{M-1}]$. |
| $A \in \mathbb{R}^{d \times d}$ | Koopman linear operator. |
| $f_\theta$ | Autoregressive NN with parameters $\theta$. |
| $p(z)$ | Equilibrium probability distribution of the latent variable $z$. |
| $s(z) = \nabla_z \log p(z)$ | Score function of the equilibrium distribution. |
| $\epsilon_\theta(z_t, \tau)$ | Pre-trained LD-FPG diffusion model (denoiser). |
| $\sigma_\tau$ | Noise schedule standard deviation from the diffusion model at step $\tau$. |
| $\Delta t$ | Integration time step for Langevin dynamics. |
| $T$ | Temperature parameter for Langevin dynamics. |
| $\eta_t$ | Stochastic noise term, $\eta_t \sim \mathcal{N}(0, \sigma_\eta^2 I)$ (scale $\sigma_\eta$ specified per experiment). |
| $d_z$ | Per-atom encoder embedding width (ChebNet channels). |
| $d_{\text{flat}}$ | Flattened pooled latent dimension after $H \times W$ pooling (here $50 \times 2 \Rightarrow d_{\text{flat}} = 100$). |
| $d$ | Shorthand for $d_{\text{flat}}$ (the flattened pooled latent dimension). |

Table 15: **Glossary of Biophysical Terms.** Definitions of key concepts from molecular dynamics and structural biology used in this work, tailored for a machine learning audience.

| Term | Description for an ML Audience |
|---|---|
| **Collective Variable (CV)** | A low-dimensional function of atomic coordinates (e.g., a distance or angle) designed to capture a specific, slow dynamic process like protein folding. Traditional simulation methods often require pre-defining good CVs; our work uses a learned latent space to discover them automatically. |
| **Potential of Mean Force (PMF)** | Essentially, an effective "energy landscape" for a molecule in solution. Lower values correspond to more probable (stable) conformations. It is the target distribution that score-guided Langevin dynamics aims to sample from, often visualized as a "free-energy surface". |
| **Dihedral Angles ($\phi, \psi, \chi$)** | Rotational angles around covalent bonds that define the geometry of a molecule.
• $\phi$ (**phi**), $\psi$ (**psi**): Define the rotation of the protein **backbone**.
• $\chi$ (**chi**): Define the rotation of the amino acid **side-chains**.
Their statistical distributions are a sensitive measure of structural fidelity. |
| **Ramachandran Plot** | A 2D plot of the backbone dihedral angles ($\phi, \psi$). Certain regions of this plot are "allowed" based on steric constraints, leading to characteristic high-probability basins that correspond to stable secondary structures like alpha-helices and beta-sheets. |
| **Rotamer** | A discrete, low-energy, and therefore highly probable conformation of a protein's side-chain, defined by its set of $\chi$ angles. A key test for generative models is whether they can reproduce the correct statistical distribution of these rotameric states. |
| **GPCR** | *G protein-coupled receptor*. A large and important family of transmembrane proteins that act as cellular signal transducers. They are highly dynamic and switch between different functional states (e.g., inactive, active), making them an ideal and challenging test system for dynamic models. |
| **lDDT** | *local Distance Difference Test*. A metric for assessing the quality of a protein structure prediction by evaluating how well local inter-atomic distances are preserved relative to a reference structure. Unlike RMSD, it is less sensitive to global rotations and more focused on local geometric accuracy. |
| **RMSF** | *Root-Mean-Square Fluctuation*. For each atom, this metric calculates the standard deviation of its position over time in a simulation trajectory. It measures the "amplitude of motion" or flexibility of different parts of the protein. |

