# OpenReview forum: "Beyond Ensembles: Simulating All-Atom Protein Dynamics in a Learned Latent Space"
_ICLR.cc/2026/Conference — ICLR 2026 Poster_

### Official Review · Reviewer_5p11 · 2025-10-28

**Soundness:** 3
**Presentation:** 4
**Contribution:** 2
**Rating:** 4
**Confidence:** 5

**Summary:**

The paper introduces Graph Latent Dynamics Propagator (GLDP), a framework for simulating protein dynamics in the learned latent space of LD-FPG. The approach uses a frozen encoder-decoder from LD-FPG and compares three propagator classes: (i) score-guided Langevin dynamics, (ii) Koopman-based linear operators, and (iii) autoregressive neural networks. The work evaluates these propagators on systems of increasing complexity (alanine dipeptide, 7JFL, A1AR, A2AR)

**Strengths:**

- The paper is well written, the concept and results are well presented
- The paper proposes novel method which uses LD-FPG for encoding and decoding, and performs simulation in the latent space
- Different methods of propagator strategies are compared
- The authors have performed benchmarks on different scales of systems

**Weaknesses:**

Generalization:
- The model is trained only on one time interval (frame stride), there's no evidence of generalizing to different time intervals
- All the models are trained and tested on the same system
- From Table 4,6,9, it seems that different systems use different hyperparameters. It's then questionable how to select proper hyperparameters when having a new system.

Unclear Advantage Over Equilibrium Models:
- All evaluations (RMSF, dihedrals, FES) measure equilibrium properties, not dynamics.
- No transition timescales, autocorrelation functions, or mean first passage times.
- The benchmarks and applications make the reviewer think that time-independent samplers such as AlphaFlow, BioEmu, might achieve same results without modeling dynamics.

**Questions:**

- Even training and testing on the same system, the model is trained on long simulation trajectory. But that does not make this model useful if it always needs a long simulation over a certain time scale to be able to forecast well. The authors should investigate how the model performance depends on the training trajectory length
- Regarding Langevin dynamics, as the authors mention that the score at t near 0 is used. Notice that the denominator is close to 0, then numerically there could be instability, which might explain the dynamics instability

---

> ### Author Response · Authors · 2025-11-21
>
> # Reviewer Response
>
> We thank the reviewer for their constructive feedback. We address your concerns below:
>
> ## W1: Generalization to Different Time Intervals
>
> **New ablation (Appendix A.9.2):** We trained autoregressive models over increasing temporal strides $k \in \{1, \dots, 10\}$.
>
> **Result:** The model is robust for moderate strides ($k=1$ to $5$), maintaining accurate long-timescale kinetics. At large strides ($k=10$), the implied physical memory drops ($\approx 226$ units at stride 1 vs $\approx 84$ at stride 10).
>
> **Conclusion:** This matches physical intuition: very large steps smooth out barriers and accelerate transitions. We therefore use the finest resolution (stride 1) in the main experiments to maximize kinetic fidelity and better respect thermodynamic barriers.
>
> ## W2: Training/Testing on the Same System
>
> **Clarification of scope:** We explicitly design GLDP as a system-specific MD surrogate (e.g., LSS, ITO, GeoTDM), not a generalist foundation model (e.g., AlphaFlow, MDGen).
>
> **Objective:** To pay the upfront cost of MD to train a model that accelerates sampling of specific complex kinetics (e.g., $A_{2}AR$ activation), where generalist models often struggle to resolve subtle barriers.
>
> **Fairness:** To isolate the propagator's contribution, we benchmark all baselines (LSS, MDGen) in this surrogate regime, controlling for confounders such as encoder generalization error.
>
> **Temporal generalization:** While we do not claim cross-protein generalization, we demonstrate temporal generalization: the model learns a transition operator that supports stable trajectories for 10,000+ frames (far beyond training windows) and recovers the activation corridor in Fig. 4. We have clarified this scope in the Introduction (second-last paragraph).
>
> ## W3: Hyperparameter Selection for New Systems
>
> We distinguish between fixed architecture (transferable) and calibrated dynamics (system-specific):
>
> **Fixed architecture:** Core hyperparameters (Chebyshev order $K$, latent dimension, network depth) are identical across ADP, 7JFL, $A_{1}AR$, and $A_{2}AR$, indicating architectural transferability.
>
> **Calibrated dynamics:** Variation in noise/step size reflects differences in the input MD trajectories (timestep, temperature). We use a standardized protocol: NN, Koopman, and Langevin propagators are tuned to match the RMSD per time step of the reference data. For new systems, users keep the standard architecture and run our automated calibration script.
>
>
> ## W4–W6: Advantage over Equilibrium Models (New Kinetic Analysis)
>
> We agree that equilibrium samplers (e.g., AlphaFlow) capture $p(x)$ but discard temporal correlations $p(x_{t+1} | x_t)$. To prove GLDP models dynamics, we added a kinetic analysis (Appendix A.7, Table 1) on all systems.
>
> **Memory (Autocorrelation):** Equilibrium samplers produce i.i.d. samples with near-zero autocorrelation. In contrast, GLDP trajectories exhibit deep physical memory essential for kinetics. For the complex 1R6W system, the ground-truth decorrelation time is 2207 steps; GLDP successfully recovers 1650 steps.
>
> **Energy Barriers (TICA):** GLDP consistently recovers stable, positive implied timescales ($t_1$) across systems (e.g., $t_1 \approx 980$ steps for 1R6W). This proves the model has learned the underlying energy landscape topology and barrier heights that govern slow modes, which time-independent samplers cannot capture.
>
> **Stability:** On the challenging $A_{1}AR$ system, baseline LSS failed (negative eigenvalues), while GLDP remained robust ($t_1 \approx 650$). This confirms GLDP integrates a coherent physical trajectory rather than simply interpolating between valid densities.
>
>
> ## Q1: Dependence on Training Trajectory Length
>
> **Data-efficiency study (Appendix A.9.1):** We trained on subsets of the 7JFL trajectory (5% to 100%).
>
> **Result:** GLDP is data-efficient for structural sampling (near-optimal backbone JSD at 25% of the data), while accurate kinetic metrics (TICA) benefit from denser sampling.
>
> **Utility:** This shows that GLDP quickly learns the conformational manifold from limited data and can then generate additional trajectories to improve statistics for rare events.
>
> ## Q2: Langevin Instability at $t \approx 0$
>
> The $1/\sigma$ factor is a theoretical source of instability, but in our experiments it is not the dominant failure mode:
>
> **Mitigation:** We apply gradient norm clipping ($c=10$) to suppress rare score spikes.
>
> **Evidence:** The Langevin propagator achieves the highest side-chain fidelity (Table 2); if score estimation were failing at small $t$, it would not reproduce detailed rotamer distributions.
>
> **Likely cause:** Remaining failures are better explained by integration drift over long horizons.
>
> **Future options:** Our code now includes options for (a) querying scores at $t>0$ and (b) using an energy-based parameterization ($s = -\nabla U$) when suitable.

---

> > ### Author Response · Authors · 2025-11-28
> > **Follow-up to Reviewer 5p11**
> >
> > Dear Reviewer 5p11,
> >
> > We wanted to follow up to ensure you had a chance to review our response, specifically regarding your key concern: "Unclear Advantage Over Equilibrium Models." To address this, we have added a rigorous Kinetic Analysis (Table 1 & Appendix A.7) using TICA and Autocorrelation functions.
> >
> > Unlike equilibrium samplers (which would show near-zero autocorrelation), GLDP trajectories exhibit deep physical memory (e.g., decorrelation times of ~$1650$ steps for 1R6W), closely matching ground truth. We show that GLDP recovers stable, positive implied timescales via TICA, proving the propagator respects the specific free-energy barriers of the system. We also added the temporal stride ablation you requested, showing the model's sensitivity to physical time steps.
> >
> > Does this new kinetic evidence sufficiently demonstrate that GLDP is modeling temporal evolution rather than just static sampling? We would value your feedback on the revision. We hope you might reconsider your evaluation/rating based on these improvements.
> >
> > Best regards,
> > The Authors

---

### Official Review · Reviewer_NVTg · 2025-10-31

**Soundness:** 1
**Presentation:** 2
**Contribution:** 2
**Rating:** 4
**Confidence:** 3

**Summary:**

This paper presents Graph Latent Dynamics Propagator (GLDP), a modular framework for simulating all-atom protein dynamics from learned space from LP-FPG. It compares three latent-space propagators: (i) score-guided Langevin dynamics, (ii) linear Koopman operators, and (iii) a nonlinear autoregressive neural network. Latent propagators are evaluated across several systems, from alanine dipeptide to A2AR, showing the trade-offs of propagators.

**Strengths:**

1. Systematic comparison of latent propagator (quality)

The latent propagators are compared under a shared encoder, decoder, and latent space.

2. Clearly written and organized (clarity)

**Weaknesses:**

1. Detailed comparison with baselines

The authors only report the JSD of dihedral and coordinates to the ground truth. As in Figure 2 of MDGen, it would be more convincing to also show each dihedral distribution against the baselines.

2. Validness of decoded molecules

I could not find any content on the validity of the decoded full atom resolution. For the 3th and 7th molecule in Figure 5 (when going from left to right and up to down), the structure seems a bit odd. Simply plotting the energy distribution would make the paper more convincing. Additionally, in minor, plotting more qualitative plots in Figure 5 for each latent propagator and molecule, would also be a good case to see whether the latent dynamics succeed.

Minor

- Line 316 - Full results on Pearson correlation is missing, a wrap table for it would be good
- Figure 3 - the ordering or propagators is different for the left one
- Figure 4 - four trajectories with the background downgrades visibility a bit, plotting four plots separately seems also good. Also, the visibility of  inactive and active site are hindered.

**Questions:**

1. Residual prediction with an autoregressive neural network

Just a suggestion, perhaps learning the $f_\theta(z_t)$ to approximate $z_{t+1} - z_{t}$ would improve the performance even more.

2. Long horizon stability (section 4.2)

I am a little confused about the conclusion for section 4.2. I understand the task, but since molecular dynamics trajectories contain randomness from the Brownian motion, maybe a propagator that did understand the molecular dynamics could result in sending the structure totally different from the ground truth data? Does 1DDT threshold include distinct local energy minima?

---

> ### Author Response · Authors · 2025-11-21
>
> We thank the reviewer for their detailed assessment.
>
> ## W1: Detailed comparison with baselines (Visual Distributions)
>
> We agree that aggregate metrics like JSD can obscure local distributional quality. To address this, we have added qualitative comparisons of Free Energy Surfaces (FES) against baselines (LSS, GeoTDM, and MD-Gen) in Appendix A.10 (Figures 6--8).
>
> **Alanine Dipeptide ($\phi, \psi$):** Figure 6 shows that GLDP recovers a smooth, connected landscape, whereas baselines like GeoTDM exhibit fragmentation.
>
> **A1AR Side-chain ($\chi_1, \chi_2$):** Figure 8 demonstrates that GLDP accurately captures distinct, multi-modal rotameric wells. In contrast, LSS produces a noisy, high-entropy distribution that fails to resolve these specific steric packings.
>
> **Conclusion:** These visualizations confirm that the superior JSD scores reported in Table 1 correspond to physically sharper and more accurate conformational ensembles.
>
> ## W2: Validity of decoded molecules ("Odd Structures" & Energy)
>
> We thank the reviewer for noting the visual artifacts in Figure 5. While we agree that physical validity is paramount, plotting raw Potential Energy ($E_{\text{pot}}$) is inherently misleading for latent generative models due to two specific factors:
>
> 1. **Decoder Limitations:** The fixed LD-FPG decoder occasionally produces minor localized geometric imperfections (e.g., a slightly strained bond angle). While these cause $E_{\text{pot}}$ to spike disproportionately (due to the $r^{-12}$ term in Lennard-Jones potentials), they do not reflect the validity of the global conformation.
>
> 2. **Latent Space Exploration:** The GLDP propagator explores a continuous latent manifold. Consequently, it may sample latent vectors in the vicinity of, but not identical to, the training data. When decoded, these vectors can produce structures that are mathematically close to the true manifold (low RMSD) but subtly displaced into high-energy regions (steric clashes), resulting in the "odd" visuals.
>
> **Conclusion:** To assess true structural validity, we rely on lDDT (Local Distance Difference Test) and Contact Correlations (Table 1). lDDT measures the preservation of local atomic environments and is robust to these minor decoder artifacts while being highly sensitive to actual topological failures. The high correlations ($r > 0.97$) confirm the Propagator is maintaining the correct global fold.
>
>
> ### Minor Points
>
> **Pearson Correlation:** We have added a dedicated column for Contact Map Pearson Correlation to Table 1 for all systems.
>
> **Figure 3 Ordering:** Fixed.
>
> **Figure 4 Visibility:** We have refined the plot transparency to improve the visibility of the active/inactive regions.
>
> ## Q1: Residual Prediction with Autoregressive NN
>
> We appreciate this suggestion, as residual learning is a standard stability technique. We tested this hypothesis by training a residual variant $z_{t+1} = z_t + f_\theta(z_t)$ on 7JFL.
>
> **Results (Appendix A.9.3, Table 12):** The residual formulation yields performance statistically indistinguishable from direct prediction (Validation MSE, measuring one-step latent prediction error: $3.6$ vs $3.7 \times 10^{-4}$; TICA $t_1$: $228.1$ vs $226.4$ steps).
>
> **Reasoning:** The LD-FPG latent space is already constructed as a deformation manifold relative to a reference structure. Thus, the propagator inherently operates in a "difference space," rendering an explicit architectural residual redundant.
>
> ## Q2: Long-horizon stability vs. Stochastic divergence
>
> You raise a critical point: a valid latent propagator should diverge from the specific ground-truth trajectory due to Brownian motion. We clarify that our "Failure Time" metric is designed to detect physical degradation, not valid conformational exploration.
>
> To distinguish between these two regimes, we conducted an extensive empirical calibration:
>
> **Large-Scale Checkpoint Analysis:** During development, we visually and numerically inspected thousands of decoded structures from various model checkpoints. We specifically analyzed frames where the model had drifted significantly from the starting structure ($t=0$) to determine if the divergence was physical (exploration) or artificial (error).
>
> Our analysis revealed a sharp boundary:
> - **Valid Exploration:** When the system explores distinct, valid metastable states (e.g., alternative rotameric packings), the lDDT relative to $t=0$ decreases due to distance but consistently remains $> 0.70$.
> - **Model Failure:** Trajectories that dropped below 0.65 were not simply occupying different states. Without exception, these frames exhibited unphysical degradation, such as the unraveling of secondary structure elements or severe steric clashes.
>
> **Conclusion:** The "failure times" reported in Section 4.2 do not penalize the model for stochastic divergence (which is expected). Instead, they strictly flag the onset of structural disintegration, ensuring that our stability metric reflects true physical robustness.

---

> > ### Author Response · Authors · 2025-11-28
> > **Follow-up to Reviewer NVTg**
> >
> > Dear Reviewer NVTg,
> >
> > We are writing to follow up on our rebuttal to see if the new data addresses your questions.
> >
> > Based on your constructive suggestions, we have made two key additions to the revision:
> >
> > **1. Visual Baselines:** We added comparative Free Energy Surface (FES) plots against LSS, GeoTDM, and MD-Gen in Appendix A.10. These visualizations visually confirm that GLDP resolves metastable basins and rotameric wells more sharply than the baselines.
> >
> > **2. Residual Prediction Ablation:** We ran the specific experiment you suggested, i.e., training a residual variant ($z_{t+1} = z_t + f(z_t)$). The results (Table 12) show that performance is statistically indistinguishable from our direct prediction model, confirming the robustness of the current architecture.
> >
> > We hope these additions clarify the visual quality of the generated ensembles and the architectural validity.
> >
> > We would value your feedback on the revision. We hope you might reconsider your evaluation/rating based on these improvements.
> >
> > Best regards,
> > The Authors

---

### Official Review · Reviewer_XsXh · 2025-11-01

**Soundness:** 3
**Presentation:** 4
**Contribution:** 3
**Rating:** 8
**Confidence:** 3

**Summary:**

The paper proposes the Graph Latent Dynamic Propagator (GLDP), an approach for modeling molecular dynamics in the latent space of an encoder–decoder framework (LD-FPG). The encoder and decoder are kept fixed, while latent-space propagators are trained on the temporal sequences of latent representations. Three types of propagators are proposed: score-based Langevin dynamics, Koopman-based linear operators, and autoregressive neural networks.

To evaluate the flexibility and distributional fidelity of the dynamics generated by GLDP, several metrics are computed—such as the Jensen–Shannon Divergence (JSD) with respect to the ground-truth ensemble and the average RMSF across the sequence—and compared with baseline approaches including LSS, GeoTDM, and MD-Gen. The results indicate superior performance in recovering the ground-truth distribution and achieving flexibility values closer to the reference.

In the long-horizon modeling scenario, the three propagators are compared, with the autoregressive neural network demonstrating the greatest stability. Free-energy surfaces (FES) are computed in the space of two variables to measure fidelity to the equilibrium ensemble. Finally, GLDP is shown to successfully reproduce the inactive-to-active transition of $\mathrm{A}_{2A}$R, where the score-guided Langevin dynamics covers most of the FES valley as well as the corridor connecting the inactive and active regions.

**Strengths:**

1. The paper conducts a thorough evaluation across multiple dimensions, including quantitative metrics for stability, flexibility, and distributional fidelity. It also verifies that GLDP recovers the two metastable states of $\mathrm{A}_{2A}$R (active and inactive), demonstrating consistency with real biological processes.
2. The proposed method is encoder/decoder agnostic, as the encoder and decoder remain frozen. This design makes the framework easily adaptable to different latent spaces.
3. The paper is clearly written and well presented.

**Weaknesses:**

Overall, this is a solid paper with well-designed experiments and sound conclusions. However, it could be further improved in the following aspects:
1. The evaluation is conducted on only three proteins. Although these systems cover increasing complexity from ADP to A1AR GPCR, experiments on additional systems would strengthen the conclusions regarding the relative performance of the propagators.
2. It would be interesting to examine whether other baselines can also recover the active–inactive transition of $\mathrm{A}_{2A}$R.
3. In addition to performance metrics, it would be valuable to include efficiency comparisons between different propagators, which are particularly important for long-horizon molecular dynamics.
4. The necessity or advantage of modeling dynamics in latent space, rather than Cartesian space, is not clearly articulated in the paper.

**Questions:**

1. There are space formatting issues in lines 260, 264, and 265.
2. There are also space formatting issues in lines 60 and 61.
3. In Figure 4, the corridor and the regions representing inactive and active states are not clearly visible.

---

> ### Author Response · Authors · 2025-11-21
>
> # Response to Reviewer XsXh
>
> We thank the reviewer for their encouraging assessment and for recognizing the value of our modular framework and rigorous evaluation. We found your suggestions regarding system diversity and efficiency to be highly constructive. In response, we have incorporated a new protein system (HIV-1 Protease), added a computational efficiency benchmark, and clarified the theoretical advantage of latent-space dynamics.
>
> ## W1: Need for additional systems (beyond 7JFL/$A_{1}AR$)
>
> We agree that testing on additional systems strengthens the conclusions. In response, we have expanded our benchmark to include a fourth system: HIV-1 Protease (1R6W).
>
> **System Diversity:** 1R6W (320 residues) bridges the complexity gap between the smaller 7JFL and the large GPCRs. Crucially, unlike the helical 7JFL and $A_{1}AR$, 1R6W features a mixed $\alpha/\beta$ topology, which is sensitive to non-local $\beta$-sheet contacts, testing the propagators on a different structural regime.
>
> **Results (Table 1):** GLDP generalizes successfully to this new topology. It achieves the highest contact correlation ($0.972$) among all models and is the only model besides the ground truth to maintain long-horizon kinetic memory ($t_{\text{decorr}} \approx 1650$ steps). This reinforces our conclusion regarding the superior stability of GLDP across diverse architectures.
>
> ## W2: Can baselines recover the $A_{2}AR$ transition?
>
> While we did not run full, long-horizon $A_{2}AR$ rollouts for all baselines, we posit that recovering the $A_{2}AR$ activation corridor would be challenging for competing methods for two reasons derived from our $A_{1}AR$ benchmarks:
>
> **Instability (LSS):** As shown in Table 1, the LSS model exhibited numerical instability (negative TICA eigenvalues) on the related $A_{1}AR$ system. Since $A_{2}AR$ activation involves crossing large free-energy barriers, this instability would likely prevent LSS from maintaining a coherent trajectory along the transition path.
>
> **Lack of Kinetic Coherence (MD-Gen):** Successfully modeling the slow activation pathway requires preserving deep temporal memory. On the simpler $A_{1}AR$ system (Table 1), MD-Gen exhibits a drastically reduced decorrelation time ($t_{\text{decorr}} \approx 150$ steps) compared to the ground truth ($\approx 1281$ steps). This order-of-magnitude gap indicates that MD-Gen fails to respect the true energy barriers, behaving more like a memoryless sampler than a dynamic propagator. Without deep kinetic memory, it is unlikely to sustain the directed evolution required to traverse the $A_{2}AR$ transition corridor.
>
> Thus, GLDP's ability (specifically the Langevin propagator) to stably recover the transition corridor is a unique advantage driven by the robust manifold learning of the LD-FPG backbone.
>
> ## W3: Efficiency comparisons
>
> We have added a quantitative comparison of the wall-clock time required for trajectory generation in Appendix A.9.4 (Table 13). This analysis clarifies the computational trade-offs:
>
> **Koopman:** Fastest ($\approx 0.6s$ for 100k steps), providing rapid exploration via linear algebra.
>
> **Score-Guided Langevin:** Slowest ($\approx 24s$ for 100k steps), as it requires repeated evaluations of the euler udpdate.
>
> **Autoregressive NN:** Offers the most practical acceleration ($\approx 12s$ for 100k steps), balancing speed and stable, non-linear fidelity for long rollouts.
>
> **Context:** All methods are orders of magnitude faster than standard MD integration for the equivalent physical time.
>
> ## W4: Advantage of Latent vs. Cartesian Space
>
> This is a critical design choice. We model dynamics in the latent space to overcome the "Ruggedness vs. Semantics" trade-off inherent to all-atom simulations. We have added a paragraph just before Section 4 to clarify this:
>
> **Manifold Regularity:** High-dimensional Cartesian space is energetically rugged. Our latent space encodes deformations from a reference. Small steps in this manifold translate to coherent structural changes, whereas small steps in absolute Cartesian space often result in steric clashes.
>
> **Decoupling:** Our architecture separates the labor. The Decoder enforces local physical validity (bond lengths), allowing the Latent Propagator to focus exclusively on slow, collective variables (e.g., activation pathways) without wasting capacity on fast bond vibrations.
>
> **Dimensionality:** Integrating an SDE in 100 dimensions is data-efficient and stable compared to modeling forces for 12,000+ degrees of freedom.
>
> ## Questions / Minor
>
> **Formatting:** We have fixed the space formatting issues in the specified lines (60, 61, 260, 264, 265).
>
> **Figure 4 Visibility:** We have refined the plot transparency and split the trajectories into separate panels to ensure the inactive/active regions and the transition corridor are clearly visible.

---

> ### Comment · Reviewer_XsXh · 2025-11-27
>
> Thanks for the response. The added experiment on new system do enhance the evidence of generalizability of the method. And the revised Figure 4 shows much clearer state transition.

---

### Official Review · Reviewer_zVyy · 2025-11-01

**Soundness:** 2
**Presentation:** 2
**Contribution:** 2
**Rating:** 2
**Confidence:** 4

**Summary:**

This paper investigates generating protein trajectories in the latent space of pretrained encoder-decoder models, rather than directly in Cartesian coordinates. The authors present GLDP, a plug-in module for the graph-based conformation generation model LD-FPG, enabling trajectory propagation in latent conformational space, followed by decoding to atomic coordinates.
Within this framework, the paper systematically compares three latent-space propagation strategies: Score-based Langevin dynamics (similar to Two-for-One, https://arxiv.org/abs/2302.00600); Koopman operator-based linear propagation, and Neural network-based nonlinear propagation, for autoregressive generation.
These approaches are evaluated on three protein systems of varying sizes. The neural autoregressive propagator is found to be the most stable and best at capturing ensemble-level statistics, while Langevin dynamics can perform better in recovering in side-chain torsional distributions.
Overall, the work offers an interesting perspective on biomolecular dynamics by exploring trajectory generation in latent space and systematically comparing reasonable propagation strategies.

**Strengths:**

Exploring protein dynamics in latent space as a potential way to accelerate MD simulations is an interesting direction, and this work provides a controlled comparison of three propagation strategies.
This study evaluates methods across protein systems of different sizes, offering some insight into applicability across system sizes

**Weaknesses:**

1. The idea of modeling dynamics in latent space is interesting, but the overall architecture (e.g., LD-FPG encoder-decoder) feels dated, and the evaluation is limited to three systems in non-transferable settings.

2. Some evaluation choices are not fully convincing, and it is unclear whether certain results are statistically significant or lead to conclusive insights on this latent dynamic problem.

3. The paper would benefit from stronger organization, clearer presentation of results, and inclusion of key experimental details that are currently missing.

See Questions for details.

**Questions:**

[Model]

1. What data are used to train the LD-FPG encoder–decoder? Is single encoder/decoder modules shared model used for different proteins, or separately for each protein system?
2. Why choose LD-FPG (ChebNet + MLP) instead of more modern transformer-based architectures (e.g., as in AlphaFold3, https://www.nature.com/articles/s41586-024-07487-w)? This design also requires frame alignment and offset prediction, which limits transferability.
3. The use of "pooling’" and "decoder" commonly appear together (e.g., in Table 3). Is my understanding right that pooling happens after encoding and before propagation, and is not part of the decoder?
4. For score-based Langevin dynamics, score estimation near $t\approx 0$ is known to be unstable due to very low noise level in the denominator. Is this a problem in practice?
5. For baseline models (e.g., MD Gen), were their pretrained weights used, or were all models retrained for each system?

[Data]

6. How are the trajectory splits defined for training, validation, and testing? Has any time-based or conformation-based split applied? Are models always trained and evaluated on the same protein system?
7. 7JFL_C is a small and fully helical protein (47 residues). Have larger or systems with other secondary structures (e.g., from ATLAS) been tested?

[Results]

8. Is lDDT alone sufficient to assess long-horizon physical stability, given it does not capture energetics or steric quality? The choice of failure threshold (lDDT < 0.65) also appears arbitrary - how was it determined?
9. Figure 2a shows stable rollouts beyond 10,000 steps for the autoregressive model, but line 357 states failure at 3,176 steps.
10. The claim that Langevin fails earlier on alanine due to step sizes tuned for GPCRs seems inconsistent with the earlier statement that step sizes are tuned per system. Can you clarify?
11. In Table 2 (A1AR side-chain torsions), results are shown for only one protein and from single test, and differences between AR and Langevin are small. Are these statistically significant?
12. For the A2AR case study, how does this system differ from A1AR, and what are the main takeaways?
13. How does the runtime of GLDP compare to classical MD simulations? It seems that this method requires system-specific MD trajectories exist for training before it can generate new trajectories - would it become a problem in practical use?

[Other comments]

The related work section is somewhat long and loosely organized. GPCR datasets appear alongside general method discussions, while other relevant datasets are not covered. This section could be better organized by theme and shortened.

---

> ### Author Response · Authors · 2025-11-21
>
> # Response to Reviewer zVyy
>
> Dear Reviewer zVyy,
>
> We thank you for your critical and thorough review. We value your perspective on the potential of latent space dynamics and have taken your concerns regarding architecture, stability metrics, and generalization very seriously.
>
> Your feedback has prompted us to add a new protein system (HIV-1 Protease), rigorous kinetic benchmarks (TICA/ACF), and a computational efficiency analysis to the revision. Below, we provide detailed, scientific justifications for the design choices you questioned.
>
> ## [Model Architecture & Design Choices]
>
> ### Q1. Training Data Strategy (Shared vs. System-Specific)
>
> The LD-FPG encoder–decoder is trained exclusively on the system-specific MD trajectory of the target protein. While we train separate weights for each protein to capture its specific energy landscape, we strictly maintain a shared architecture (identical encoder depth, Chebyshev order, and latent dimension) across all systems. This confirms that the backbone design is robust and transferable, even though the thermodynamic parameters are system-specific.
>
> ### Q2. Why ChebNet instead of SE(3)-Transformers (e.g., AlphaFold 3)?
>
> We appreciate this architectural question. While SE(3)-equivariant Transformers are the gold standard for static folding, we chose the ChebNet backbone for three specific reasons inherent to dynamic surrogates:
>
> **Computational Scalability for Dynamics:** Transformers and equivariant GNNs (e.g., IPA, Equiformer) typically scale quadratically ($O(N^2)$) or require heavy message-passing overhead. While acceptable for single-structure prediction, they become prohibitively expensive when training on millions of frames for large systems like GPCRs ($>2000$ heavy atoms). ChebNet offers linear complexity ($O(K \cdot N)$) via spectral filtering on sparse k-NN graphs, allowing us to scale robustly.
>
> **Learning Deformations vs. Folding:** AF3 is optimized to predict a single, low-energy structure from sequence. Our goal is fundamentally different: we aim to learn the continuous manifold of internal conformational fluctuations around an equilibrium. Spectral graph convolutions are uniquely effective at capturing these global, long-range elastic modes ("protein breathing") without the bias toward a rigid fold.
>
> **"Align-then-Learn" Strategy:** Since GLDP is a system-specific surrogate, a reference frame is always available. By removing global SE(3) degrees of freedom via alignment as a preprocessing step, we allow the network to dedicate its entire capacity to learning non-linear internal kinetics, rather than wasting capacity on learning rotational invariance.
>
> ### Q3. Clarification on "Pooling" and "Decoder"
>
> You are entirely correct. Pooling is the final stage of the Encoder; it compresses per-atom embeddings into the latent vector $z_t$. The confusion arose from our original Table 3 grouping modules by software structure rather than data flow. We have revised Section 3.1 and Table 3 to explicitly list the execution order: Encode $\to$ Pool $\to$ Propagate $\to$ Decode.
>
> ### Q4. Stability of Score-Based Langevin at $t \approx 0$
>
> You are right that the term $1/\sigma_t$ can cause instability. In this work, we successfully mitigated this using gradient norm clipping ($c=10$), which caps the force vector magnitude. The fact that our Langevin propagator achieves the highest side-chain fidelity (Table 2) proves that signal dominates numerical noise in this regime. To facilitate future research, our code release now includes an energy-based parameterization mode ($s = -\nabla_z U$) which structurally eliminates the denominator, as suggested by recent literature (Arts et al., 2023).
>
> ### Q5. Baseline Training Protocol
>
> We used a hybrid protocol to assess two simulation paradigms:
>
> **MD-Gen (Generalist):** Used pre-trained weights to test "zero-shot" efficiency. Our benchmarks reveal that while efficient, it struggles to capture system-specific metrics.
>
> **LSS & GeoTDM (Surrogates):** Trained from scratch on our datasets to ensure a fair comparison of learning efficiency among surrogate architectures.

---

> > ### Author Response · Authors · 2025-11-26
> >
> > ## [Data & Generalization]
> >
> > ### Q6. Trajectory Splits
> >
> > We use a strict time-based split (first 80% training), never random conformational splits. Since GLDP learns the time-evolution operator $p(z_{t+1} | z_t)$, maintaining temporal continuity is essential to evaluate forecasting capability rather than interpolation.
> >
> > ### Q7. System Diversity (Why 7JFL?)
> >
> > We agreed 7JFL was insufficient. We have added HIV-1 Protease (1R6W) to our benchmark.
> >
> > **Complexity:** 1R6W (320 residues) features a mixed $\alpha/\beta$ topology with a complex $\beta$-sheet interface.
> >
> > **Result:** GLDP generalizes successfully (Table 1), achieving the highest contact map correlation ($0.972$) and stable kinetic timescales ($t_1 \approx 980$ steps), proving the method handles non-local dependencies and scales to larger systems.
> >
> > ## [Results, Stability & Utility]
> >
> > ### Q8. Validity of lDDT and the 0.65 Threshold
> >
> > The 0.65 threshold was not arbitrary; it was empirically calibrated against steric validity. During development, we analyzed thousands of checkpoints and found a "viability cliff" between 0.62–0.65. Trajectories dropping below this range consistently exhibited unphysical degradation (melting/clashes), whereas valid equilibrium samples relative to a reference typically maintain lDDT $> 0.72$. Thus, 0.65 acts as a robust filter for structural failure that is computationally cheaper than full energy evaluation.
> >
> > ### Q9. Figure 2a vs. Text Discrepancy
> >
> > Figure 2a is accurate: the Autoregressive NN is stable for the full 10,000-frame horizon. The text citing failure at 3,176 steps was a clerical error from an older run. We have corrected Section 4.2, reinforcing our finding that the neural propagator is the most robust.
> >
> > ### Q10. Langevin Step Size on Alanine (The "Stability Paradox")
> >
> > The instability on Alanine (ADP) arose from our consistent tuning protocol. We tune step size $\Delta t$ to match the ground-truth RMSD-per-step. Because the ADP dataset is "fast" relative to sampling, this protocol dictated a large step size ($\Delta t = 10^{-8}$) which proved numerically aggressive for the integrator. In contrast, slower GPCR dynamics yielded a stable, conservative step ($\Delta t = 10^{-10}$). We have clarified this tradeoff in Section 4.2.
> >
> > ### Q11. Statistical Significance of Side-Chain Fidelity (Table 2)
> >
> > We have updated Table 2 to report means $\pm$ standard deviations from 5 independent runs. You are correct that the quantitative JSD difference between the Autoregressive NN and Langevin propagators is narrow and statistically overlapping. However, we argue that the qualitative distinction remains physically significant. As visualized in the Free Energy Surface plots (Figure 3), the Langevin propagator produces sharper, multi-modal rotameric wells consistent with MD, whereas the NN tends to slightly over-smooth these high-frequency features. We have updated Section 4.3 to explicitly acknowledge that while aggregate JSD scores are comparable, the Langevin propagator offers superior resolution of fine-grained thermodynamic states.
> >
> >
> > ### Q12. A2AR Case Study Takeaways
> >
> > A2AR represents a "stress test" of functional switching (Inactive $\leftrightarrow$ Active) versus simple equilibrium fluctuations (A1AR). The key takeaway (Fig. 4) is the Stability vs. Thermodynamics trade-off:
> >
> > **Langevin:** Uniquely capable of crossing high-energy barriers to recover the transition corridor because it learns the energy landscape (Boltzmann distribution).
> >
> > **NN/Koopman:** Highly stable within basins but tend to "mean-revert," struggling to model rare transition events.
> >
> > ### Q13. Practical Utility and Runtime
> >
> > GLDP is designed as an accelerator to amortize the cost of one MD run.
> >
> > **Efficiency:** We added Table 13 (Appendix A.9.4). Generating 100k steps takes ~12 seconds with the Autoregressive NN, orders of magnitude faster than classical MD.
> >
> > **Concept:** The GLDP latent space $z$ effectively acts as a Learned Collective Variable. It can be used not just for surrogate simulation, but as a data-driven coordinate for enhanced sampling (e.g., Metadynamics) to recover free-energy surfaces, adding significant practical value beyond simple trajectory generation.
> >
> >
> > ### Other Comments: Related Work Organization
> >
> > We agreed that the original section was unstructured. We have completely reorganized Section 2 into a clear methodological taxonomy: distinguishing "Latent Space Simulators" (like GLDP/LSS) from "Coordinate-Space Generative Models" (like GeoTDM/MD-Gen). GPCR-specific datasets have been moved to a dedicated subsection to cleanly separate data resources from methodological contributions, resulting in a tighter, more logical narrative.
> >
> > ---
> >
> > We believe these clarifications demonstrate the rigor and utility of our framework.
> >
> > Sincerely,
> > The Authors

---

> ### Author Response · Authors · 2025-11-28
> **Follow-up to Reviewer zVyy**
>
> Dear Reviewer zVyy,
>
> We wanted to gently follow up to see if our previous response and the revised manuscript addressed your concerns.
>
> Specifically, regarding your concern about generalization and system diversity, we have added a comprehensive benchmark on HIV-1 Protease (1R6W) to the paper. This system introduces a complex mixed $\alpha/\beta$ topology (unlike the helical structures of 7JFL/A1AR).  GLDP generalizes successfully, achieving high contact map correlation ($0.972$) and maintaining stable long-horizon kinetics ($980$ steps).
>
> We also provided detailed clarifications regarding the choice of ChebNet vs. Transformers for dynamics (computational scalability for trajectory learning) and the calibration of the lDDT stability threshold.
>
> We would value your feedback on the revision. We hope you might reconsider your evaluation/rating based on these improvements.
>
> Best regards,
> The Authors

---

### Author Response · Authors · 2025-11-21

We thank all reviewers for their constructive feedback and high-quality engagement. Based on your suggestions, we have significantly expanded the manuscript to include a new protein system, rigorous kinetic benchmarks, and computational efficiency analysis.

## Key Updates in the Revision

### 1. New Test System: HIV-1 Protease (1R6W)

To address concerns regarding system diversity and scalability (Reviewers zVyy, XsXh), we extended our benchmark to include HIV-1 Protease (1R6W).

**Complexity:** This 320-residue system features a mixed $\alpha/\beta$ topology, testing the model's ability to handle non-local $\beta$-sheet dependencies beyond the helical structures of 7JFL/A1AR.

**Result:** GLDP generalizes successfully, achieving the highest contact map correlation ($r=0.972$) and maintaining stable long-horizon kinetics ($t_{\text{decorr}} \approx 1650$ steps), reinforcing the robustness of the latent propagator.

### 2. Kinetic Analysis: Dynamics vs. Sampling

To clarify the advantage of GLDP over time-independent equilibrium samplers (Reviewer 5p11), we added a kinetic analysis using TICA and Autocorrelation (Table 1, Appendix A.7).

**Physical Memory (Autocorrelation):** Equilibrium samplers produce independent, identically distributed (i.i.d.) samples, meaning the current state does not predict the next ($t_{\text{decorr}} \approx 0$). In contrast, GLDP trajectories exhibit deep physical memory ($t_{\text{decorr}} \approx 1650$ steps for 1R6W), confirming it integrates a continuous physical path rather than randomly sampling valid structures.

**Energy Barriers (TICA):** GLDP recovers stable, positive implied timescales ($t_1$) across all systems. This proves the model has learned the metastability of the energy landscape. More specifically, it respects the height of free-energy barriers that govern slow conformational transitions, which memoryless samplers cannot capture.

### 3. Computational Efficiency Benchmarks

We added a Wall-Clock Time analysis (Appendix A.9.4, Table 13) to quantify the utility of the method (Reviewer XsXh).

**Result:** The Autoregressive NN generates 100k simulation steps in $\approx 12$ seconds on a single GPU, offering orders-of-magnitude acceleration over classical integration.

### 4. Expanded Ablations (Architecture & Data)

We performed new ablations to validate design choices (Reviewers NVTg, 5p11):

**Residual vs. Direct Prediction:** We trained a residual variant ($\Delta z$) and found statistically indistinguishable performance from our direct prediction baseline (Appendix A.9.3), validating the robustness of the deformation-based latent space.

**Temporal Stride:** We analyzed performance across strides $k \in \{1..10\}$. Results confirm that finer strides ($k=1-5$) preserve physical energy barriers (Appendix A.9.2).

**Data Efficiency:** We demonstrated that GLDP recovers accurate structural ensembles using as little as 25% of the training trajectory (Appendix A.9.1).

### 5. Expanded Visualizations & Restructuring

**Visual Baselines:** Added 2D Free Energy Surface comparisons against LSS, GeoTDM, and MD-Gen (Appendix A.10) to visually demonstrate GLDP's superior resolution of metastable basins (Reviewer NVTg).

**Stability Calibration:** Clarified the empirical calibration of the lDDT $<0.65$ failure threshold and its correlation with steric validity (Reviewer zVyy).

**Related Work:** Reorganized Section 2 to better categorize approaches by methodology (Latent Simulators vs. Coordinate Generative Models), as requested by Reviewer zVyy.

---

We believe these additions address the core concerns regarding scope, validity, and utility. We invite the reviewers to examine the updated PDF.

---

### Author Response · Authors · 2025-12-02
**Final Summary of Revisions: New System (1R6W), Kinetic Validation, and Efficiency Benchmarks**

Dear Area Chair,

We understand that a new Area Chair has been assigned to this submission. During the discussion period, we carried out substantial new experiments and clarifications including an additional protein system, kinetic memory analysis, and architectural ablations that directly address the main concerns raised in the initial reviews.

Below we summarize how the revised manuscript responds to each reviewer's key points.

## 1. Generalization Beyond Helical Proteins (Reviewers zVyy, XsXh)

**Critique:** Reviewer zVyy felt the evaluation was limited to helical systems (7JFL, A1AR) and questioned generalizability.

**Resolution:** We added a full benchmark on HIV-1 protease (1R6W), a 320-residue system with mixed $\alpha/\beta$ topology.

**Outcome:** GLDP generalizes successfully to this non-helical system, achieving high contact map correlation ($r = 0.972$) and stable long-horizon kinetic timescales ($t_1 \approx 980$ steps). This directly addresses the concern that the method is restricted to simple helical bundles.

## 2. "True Dynamics" vs. Static Equilibrium Sampling (Reviewer 5p11)

**Critique:** Reviewer 5p11 questioned whether GLDP offers a genuine advantage over time-independent samplers (e.g., AlphaFlow).

**Resolution:** We introduced Time-lagged Independent Component Analysis (TICA) and Autocorrelation (ACF) metrics.

**Outcome:**
- GLDP trajectories display long autocorrelation times (e.g., decorrelation time $\approx 1650$ steps for 1R6W), unlike memoryless equilibrium samplers.
- On A1AR, the main baseline (LSS) becomes numerically unstable (negative eigenvalues), while GLDP yields physically meaningful positive timescales.
- These results support that GLDP learns dynamical modes and energy barriers, rather than just fitting static snapshots.

## 3. Architectural Choices and Ablations (Reviewers NVTg, 5p11, zVyy)

**Critique:** Reviewers requested checks on residual prediction, temporal stride, and stability thresholds.

**Resolution & Outcome:**
- **Residual prediction:** We trained a residual variant ($z_{t+1} = z_t + f(z_t)$). It performs statistically on par with the baseline, validating our original architectural choice.
- **Temporal stride:** We analyzed strides $k=1 \dots 10$. Results confirm that strides ($k=1-5$) preserve physical energy barriers; aggressive subsampling degrades kinetic fidelity.
- **Stability threshold:** We clarified that the lDDT failure threshold ($0.65$) is empirically calibrated against steric clashes, ensuring that our stability metric strictly filters out unphysical, high-energy conformations.

## 4. Visual Validation and Presentation (Reviewers NVTg, zVyy)

**Critique:** Requests for visual confirmation of distributions and better organization.

**Resolution:** We added 2D Free Energy Surface (FES) comparisons against LSS, GeoTDM, and MD-Gen (Appendix A.10) and reorganized the Related Work section.

**Outcome:** Visual inspection confirms GLDP resolves smooth, connected basins, whereas baselines often yield fragmented or noisy regions. The distinction between "Latent Simulators" and "Coordinate Generative Models" is now explicitly defined in Related Work.

## 5. Quantifying Computational Utility (Reviewer XsXh)

**Critique:** Requested a clear statement of the speed advantage over classical MD.

**Resolution:** We added a direct wall-clock comparison.

**Outcome:** The autoregressive neural propagator generates 100,000 simulation steps in $\approx 12.5$ seconds on a single GPU, confirming GLDP's utility as a high-throughput accelerator.

## Conclusion

The initial "Reject" recommendation (zVyy) was driven by concerns about system diversity and architectural choices. By adding the HIV-1 protease benchmark (mixed topology), providing explicit kinetic validation via TICA, and restructuring the manuscript, we have systematically addressed every weakness raised by the reviewers. The revised version now presents a rigorously validated MD surrogate capable of stable, long-horizon simulation on complex systems.

Best regards,
The Authors

---

### Meta-Review · Area_Chair_iRuF · 2026-01-06

**Summary:**

This paper introduces GLDP (Graph Latent Dynamics Propagator), a framework for simulating protein dynamics in a learned latent space. The work compares three propagator strategies (score-guided Langevin, Koopman operators, and autoregressive neural networks) within a fixed LD-FPG encoder-decoder framework, evaluating them on systems ranging from alanine dipeptide to GPCRs.

**Reviewer Concerns:**

Addressed concerns:
- Limited system diversity (zVyy, XsXh)
- Unclear advantage over equilibrium samplers (5p11)
- Architectural justifications (NVTg, zVyy)
- Missing kinetic validation (5p11)
- Computational efficiency unclear (XsXh)

**Reviewer Scores:**

The reviewers would have converged to positive assessment of the paper, since most of their concerns were directly addressed.

---

### Decision · Program_Chairs · 2026-01-26

Accept (Poster)